# Mutations associated with human neural tube defects display disrupted planar cell polarity in *Drosophila*

**Ashley C Humphries**[1,2,3]**, Sonali Narang**[1,2,3†‡]**, Marek Mlodzik**[1,2,3]*****

[1]Department of Cell, Developmental and Regenerative Biology, New York, United States; [2]Icahn School of Medicine at Mount Sinai, New York, United States; [3]Graduate School of Biomedical Sciences, New York, United States

**Abstract** Planar cell polarity (PCP) and neural tube defects (NTDs) are linked, with a subset of NTD patients found to harbor mutations in PCP genes, but there is limited data on whether these mutations disrupt PCP signaling in vivo. The core PCP gene *Van Gogh* (*Vang*), *Vangl1/2* in mammals, is the most specific for PCP. We thus addressed potential causality of NTD-associated *Vangl1/2* mutations, from either mouse or human patients, in *Drosophila* allowing intricate analysis of the PCP pathway. Introducing the respective mammalian mutations into *Drosophila Vang* revealed defective phenotypic and functional behaviors, with changes to Vang localization, post-translational modification, and mechanistic function, such as its ability to interact with PCP effectors. Our findings provide mechanistic insight into how different mammalian mutations contribute to developmental disorders and strengthen the link between PCP and NTD. Importantly, analyses of the human mutations revealed that each is a causative factor for the associated NTD.

**\*For correspondence:**
marek.mlodzik@mssm.edu

**Present address:** [†]Department of Pathology, NYU School of Medicine, New York, United States; [‡]Laura and Isaac Perlmutter Cancer Center, NYU School of Medicine, New York, United States

**Competing interests:** The authors declare that no competing interests exist.

## Introduction

Neural tube closure defects (NTDs) are common congenital malformations in humans, affecting approximately 1 in 1000 births. During embryogenesis, the neural plate undergoes shaping, before neural folds develop and elevate, contact, and then fuse at the midline to create the closed tube structure that will form the spinal cord and brain (*Copp et al., 2003*). Studies in *Xenopus*, zebrafish, and subsequently mouse, revealed a role for planar cell polarity (PCP) signaling in this process. Further, it was demonstrated that *Looptail* (*Lp*) mice, with mutations in the *mVangl2* locus (see below), fail to establish a neural plate width that will allow for bending and closure (*Etheridge et al., 2008*; *Wallingford and Harland, 2002*; *Wang et al., 2006a*; *Ybot-Gonzalez et al., 2007*). Consistent with a role in neurulation, *Vangl2* (an orthologue of the *Drosophila* PCP core gene *Van Gogh/Vang*) is expressed broadly in the neural plate prior to, during, and after closure (*Kibar et al., 2001*). Interestingly, the second orthologous mouse gene, *Vangl1*, is also expressed in this tissue; however, its expression is more restricted, showing a complementary pattern with *Vangl2* (*Doudney et al., 2005*; *Torban et al., 2008*; *Torban et al., 2007*).

*Lp* was originally defined by two independent mutations (D255E and S464N) in *mVangl2* that map to the C-terminal tail. Both display an identical phenotype, with homozygous mice developing craniorachischisis, a completely open neural tube (*Kibar et al., 2001*; *Murdoch et al., 2001*). At the molecular level, the mutations were found to disrupt binding between Vangl2 and effectors and also showed reduced membrane localization, decreased protein levels, and defective phosphorylation of Vangl2 (*Devenport and Fuchs, 2008*; *Gao et al., 2011*; *Guyot et al., 2011*; *Murdoch et al., 2014*; *Song et al., 2010*; *Torban et al., 2004*; *Torban et al., 2007*). Overall, this led to suggestions that the mutants are loss-of-function (LOF), and that Vangl2 is a rate-limiting component in a dosage-sensitive pathway. However, this is in contrast to the earliest genetic studies, which suggested a semi-

**eLife digest** As an embryo develops, its cells must work together to build mature tissues and organs. During the formation of the nervous system, for example, a sheet of cells destined to become the brain and spinal cord folds up into a tube spanning the length of the embryo. Normally, this tube – known as the 'neural tube' – zips up, and the cells that will eventually become skin and other surrounding tissues close in over it.

If the neural tube does not close completely, different parts of the spinal cord or brain can remain unprotected. This can cause diseases called neural tube defects, such as spina bifida, which is characterized by holes in the backbone exposing the spinal cord and surrounding membranes. Patients with neural tube defects can have similar genetic mutations, for example, in the genes controlling a process called "planar cell polarity", or PCP for short.

Cells arranged in flat sheets use the PCP process to sense direction, and it is this process that allows structures, such as the scales on a fish or the hairs on a mouse, to all point in the same direction. PCP is also important in embryonic development: sheets of cells that can sense direction correctly can therefore move collectively to complete complex tasks (such as closing the neural tube). However, no-one knew whether the specific PCP gene mutations implicated in neural tube defects in humans actually affected the cells' ability to sense direction, or indeed whether they were even involved in causing the diseases.

Humphries et al. set out to find out more about these mutations using fruit flies as a model system. The fruit fly is widely used to study the genes and signals involved in direction sensing, especially PCP. Problems with PCP produce easily measurable changes in the wing and eye, showing what went wrong and how badly.

Humphries et al. genetically engineered fruit flies to have the same mutations as human patients and revealed that these mutations did indeed alter cells' ability to sense direction. These experiments also showed that each mutation did so in a different way, and with varying severity. This explained why the same mutations caused different levels of neural defects in mice (which are commonly used to study human diseases) and suggests that they might contribute to neural tube disorders in humans.

These results show potential connections between neural tube defects and direction sensing in cells. In the future, this study and follow-up work could help researchers to understand what types of mutation have the most impact, which may eventually allow doctors to better predict who is most at risk of being affected by these conditions.

dominant mutation with incomplete penetrance. This was due to the presence of a looped tail phenotype in the majority of, but not all heterozygotes, caused by delay of posterior neuropore closure (*Copp et al., 1994*; *Strong and Hollander, 1949*). Since the original identification of the $Vangl2^{Lp}$ alleles, other PCP mutant mice were found to develop NTDs, implicating the pathway defects as a disease risk factor (*Andre et al., 2012*; *Curtin et al., 2003*; *Etheridge et al., 2008*; *Hamblet et al., 2002*; *Merte et al., 2010*; *Murdoch et al., 2003*; *Wang et al., 2006a*; *Wang et al., 2006b*). Due to this strong correlation, efforts began to investigate whether human NTD patients might also exhibit mutations in PCP components.

A number of mutations were identified in patient populations in *VANGL1* and *VANGL2*, as well as additional core PCP and associated proteins (reviewed in *Juriloff and Harris, 2012*). In each case, the mutation was heterozygous and, interestingly, familial mutations could be found to result in different disease severity among family members. However, this is perhaps not surprising given the complex etiology of NTDs, which are thought to result from a mixture of genetic and environmental factors. Furthermore, digenic mutations were also discovered among PCP genes, consistent with a multifactorial hypothesis, whereby additional factors would be required to achieve a threshold for disease progression (*Allache et al., 2012*; *Beaumont et al., 2019*; *Chen et al., 2018b*; *De Marco et al., 2012*; *Merello et al., 2015*; *Wang et al., 2018*). This is additionally supported by chimeric mouse studies using the *Lp* allele that showed an all-or-nothing effect in developing craniorachischisis (*Musci and Mullen, 1990*).

Despite the progress in identifying mutations within PCP components, data is lacking as to whether these mutations are in fact pathological, with only a few studies addressing this question to date (*Iliescu et al., 2014*; *Kibar et al., 2007*; *Lei et al., 2010*; *Reynolds et al., 2010*). Furthermore, evidence has emerged that the *mVangl2^{LP}* allele may in fact be a dominant negative mutation (*Song et al., 2010*; *Yin et al., 2012*), indicating there is more to investigate surrounding its molecular behavior. In addition, a recessive *mVangl2^{LP}* mutation was discovered (with substitution R259L) that gave an apparently normal phenotype in heterozygotes. In this case, only 47% of homozygote animals displayed a looped tail and 12% developed spina bifida, a milder NTD as compared to craniorachischisis, leading to the suggestion that this *Lp* mutation is a hypomorphic *Vangl2* LOF allele (*Guyot et al., 2011*; *Wang et al., 2006b*; *Yin et al., 2012*).

Polarization of epithelial cells, and cells in general, is critical for the morphogenesis and function of mature tissues, with perturbation of cellular polarity and tissue organization implicated in numerous diseases. Epithelial cell polarity can be derived in two axes, apical-basal and orthogonal to the plane of the epithelium, which is referred to as planar cell polarity (PCP). PCP establishment is governed by members of the conserved non-canonical Wnt/Frizzled-PCP pathway. Besides the four-pass trans-membrane protein Vang (Vangl1 and Vangl2 in mammals, see above), which was - like all other core PCP factors - originally discovered in *Drosophila* (*Taylor et al., 1998*), a.k.a. *strabismus/stbm* (*Wolff and Rubin, 1998*), they include the atypical cadherin Flamingo (Fmi; Celsr in mammals), the seven-pass transmembrane protein Frizzled (Fz; Fzd in vertebrates with several family members), and the cytoplasmic proteins Dishevelled (Dsh; Dvl in mammals), Diego (Dgo; Inversin/Diversin in vertebrates), and Prickle (Pk). The pathway is synonymous with the asymmetric localization of these core members, which form into two sub-complexes on opposite sides of a given cell, creating an intracellular bridge to convey polarity across the tissue. The complexes also direct spatially restricted downstream signaling through tissue-specific effectors, leading to cytoskeletal rearrangement in the majority of cases (*Adler, 2012*; *Goodrich and Strutt, 2011*; *Humphries and Mlodzik, 2018*; *Singh and Mlodzik, 2012*; *Vladar et al., 2009*; *Yang and Mlodzik, 2015*).

Asymmetric localization of PCP complexes is directly observable in cells of the *Drosophila* wing where they align to the proximal-distal axis. Molecular interactions promote the formation of stable complexes at proximal (Fmi-Vang-Pk) and distal (Fmi-Fz-Dsh-Dgo) cell membranes, with Fmi forming a homotypic interaction across cells. In the wing, signaling leads to the formation of a single-actin rich hair at the distal vertex of each cell. While in other tissues displaying visible PCP features, phenotypes can include fate specification or coordinated cell movement (*Adler, 2012*; *Goodrich and Strutt, 2011*; *Humphries and Mlodzik, 2018*; *Singh and Mlodzik, 2012*). For example, in the *Drosophila* eye, PCP signaling is responsible for determining the differential fate specification of photoreceptors R3 and R4, which also directs rotation and subsequent orientation of the photoreceptor clusters, referred to as ommatidia. These signaling events in turn lead to a mirror-image arrangement of ommatidia across the dorso-ventral midline, or equator. Disruption of PCP in the wing leads to misorientation of cellular hairs or multiple wing hairs (*Adler, 2012*; *Goodrich and Strutt, 2011*; *Humphries and Mlodzik, 2018*; *Singh and Mlodzik, 2012*), while in the eye chirality or rotation defects of the ommatidia are observed (*Mlodzik, 1999*; *Strutt and Strutt, 1999*).

The importance of PCP during vertebrate development and disease has also become widely recognized (*Goodrich and Strutt, 2011*; *Humphries and Mlodzik, 2018*; *Simons and Mlodzik, 2008*; *Wang and Nathans, 2007*). The PCP pathway is highly conserved in vertebrates, with different tissues displaying PCP readouts or defects upon misregulation (*Goodrich and Strutt, 2011*; *Wang and Nathans, 2007*; *Yang and Mlodzik, 2015*). One example of this is the above mentioned *mVangl2^{LP}*, and the first example of a link between PCP signaling and NTDs (*Kibar et al., 2001*; *Murdoch et al., 2001*). Based on the fact that several human patient-derived *VANGL1/2* mutations exist, and the availability of comparable mouse alleles, we set out to investigate how these alleles impact PCP signaling functionally in a uniform defined genetic background. For this we utilized *Drosophila*. Overall, we focused on six mutations located in the C-terminal tail of Vang, where all molecular interactions have been thus far mapped. We observed that all resulted in aberrant PCP signaling and phenotypes, however, to varying degrees. In the majority of cases, the mutation was dominant, either hypermorphic gain-of-function or dominant negative, leading to altered protein localization of both alleles in vivo. We were also able to show that one mutation was indeed hypomorphic in its behavior. Taken together, our analyses demonstrate in molecular detail the nature of

the different *VANGL1/2* alleles. These alleles display defective PCP signaling in vivo, indicating their causative association with the NTD phenotypes of the human patient mutations.

## Results

### Mutations associated with mammalian NTD can be found throughout the C-terminal tail of *Vangl* genes

The mammalian core PCP genes *Vangl1/2*, and *Vang* in *Drosophila*, play essential roles in planar polarity signaling. *Vang* interacts genetically with each PCP core gene. It has further been demonstrated that it also can physically interact with all other members of the PCP complex, and the interaction sites with the cytoplasmic PCP core factors have been mapped to its C-terminal domain (*Bastock et al., 2003*; *Darken et al., 2002*; *Das et al., 2004*; *Humphries and Mlodzik, 2018*; *Jenny et al., 2005*).

While mutations have been observed along the entire span of the *VANGL1/2* genes in human NTD patients, mutations known to be causative for NTD in the mouse all map to the C-terminal tail (*Chen et al., 2013*; *El-Hassan et al., 2018*; *Guyot et al., 2011*; *Kibar et al., 2001*; *Murdoch et al., 2001*). We were interested to explore whether the different mutations observed in human patients could impact PCP phenotypically. Furthermore, we set out to investigate how mutations found in both the mouse and human patients affected signaling at a molecular and functional level. For this, we turned to *Drosophila* as it allows for intricate and quantitative analyses, and is a genetically simpler system with less redundancy - there is only one *Vang* gene in flies - that, importantly, shows functional and mechanistic conservation (*Adler, 2002*; *Adler, 2012*; *Goodrich and Strutt, 2011*; *Singh and Mlodzik, 2012*). We chose to focus our attention on mutations within the C-terminal tail due to the functional importance of this region.

A number of C-terminal mutations have been discovered to date in both mice and humans and are detailed in *Figure 1—source data 1*. For our study, we selected two mutations from the mouse and four mutations from human patients for further analysis, these include D255E and R259L identified in mouse *Vangl2* (*Guyot et al., 2011*; *Kibar et al., 2001*), M328T and R517H found in human *VANGL1* (*Kibar et al., 2007*; *Merello et al., 2015*), and R270H and R353C from human *VANGL2* (*Kibar et al., 2011*; *Lei et al., 2010*; *Figure 1*). These mutations were selected for varying reasons. Firstly, we were interested to compare mouse D255E and R259L in a genetically uniform background, as the mutations are only a few residues apart, and strikingly the more conservative mutation (D255E) displays a more severe phenotype as compared to R259L in mouse models (*Guyot et al., 2011*; *Kibar et al., 2001*). Secondly, R270H was selected, as the residue is absolutely conserved across all species, and it covers two independent mutational events, as a mutation in *VANGL1* was also observed at the equivalent residue (see *Figure 1B*) in a digenic combination along with a *CELSR1* mutation (*Chen et al., 2018a*; *Kibar et al., 2007*). Thirdly, M328T was chosen because information to date suggests this mutation may be a LOF allele (*Reynolds et al., 2010*), and R353C as it was suggested that effector binding may be reduced (*Lei et al., 2010*; *Figure 1B*). Finally, we selected R517H, a highly conserved residue with a conservative mutation where the bioinformatic prediction of its tolerance was unclear (*Figure 1—source data 1*). Overall, we analyzed mutations from both *VANGL1* and −2, investigated mutations spread along the length of the C-terminal tail (*Figure 1A*), compared conservative as well as drastic substitutions, and focused on well conserved sites across species.

### *Drosophila* provides a sensitive system for investigating the effect of mutations on PCP

To compare mutations in vivo in *Drosophila*, we took advantage of the Gal4/UAS system to allow for regulated expression of our transgenes (*Brand and Perrimon, 1993*). We also included a Flagx3-tag and an attB (bacterial attachment) site. To enable site-specific recombination, constructs were then introduced into a fly strain containing an attP (phage attachment) docking site, along with PhiC31 integrase activity to mediate integration between the attB and P sites (*Bischof et al., 2013*; *Bischof et al., 2007*). Thus, by using the same attP docking site, we ensured insertion of the transgenes at a constant genetic locus and thus equivalent expression for accurate comparison. We chose to introduce mutations into the *Drosophila Vang* gene, rather than use mouse *Vangl2* or human

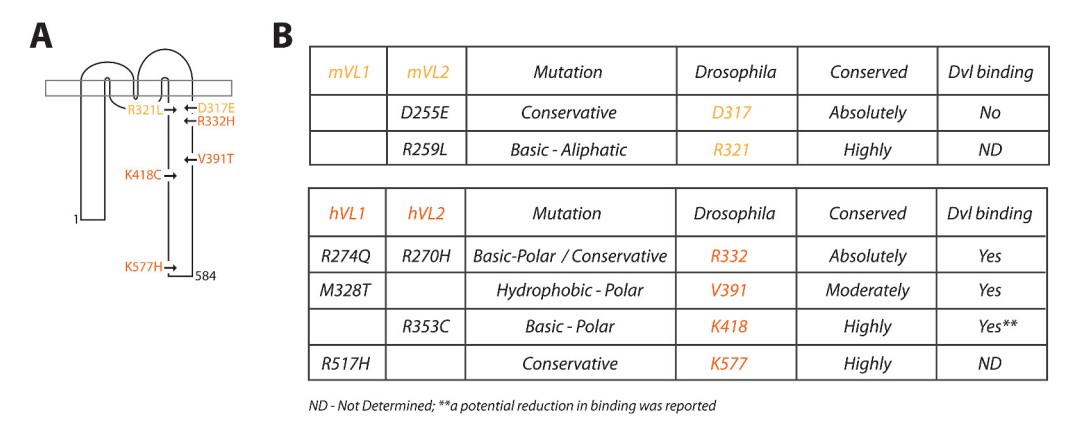

**Figure 1.** Description of Vang mutations. (**A**) Schematic showing the location of investigated mutations along the C-terminal tail of Vang. Mutations highlighted in yellow were originally identified in the mouse, mutations highlighted in orange were found in human NTD patients (see supplement for references). (**B**) Table displaying information concerning the selected mutations from mouse (top) and human patients (below). Details include: original mammalian mutation, nature of the mutational change (e.g. D to E is listed as conservative as both are acidic residues), equivalent residue in *Drosophila* Vang, whether the mutated residue is conserved among species, and whether the respective mutated Vangl protein was able to retain binding to the effector protein Dishevelled (Dvl).

The online version of this article includes the following source data for figure 1:

**Source data 1.** Summary of NTD-associated mammalian mutations found to date within the C-terminal tail of *Vangl/VANGL* genes.

*VANGL1/2*, to preserve native signaling and to reduce genetic complexity. As the equivalent residues have different numbers in *Vang*, as compared to the mouse or human orthologues, the mutations we investigated are referred by their numbers in *Drosophila*, here and throughout the paper, and are as follows: D317E, R321L, R332H, V391T, K418C and K577H (see *Figure 1B* for comparison to mammalian residue numbers).

In order to first test the contribution of individual mutations, we performed an overexpression experiment. Driving expression of wild-type (wt-)Vang with *actin-Gal4* (*ac >WT*) led to a moderate hair reorientation phenotype (*Figure 2C and D*). This allowed us to compare whether mutations showed a gain- or loss-of-function phenotype with respect to the positive (wt-Vang) and negative (*Vang*[-/-]) control. A change or an increase in phenotype quality or strength as compared to wt-Vang overexpression could suggest a dominant negative effect or a gain-of-function in activity, while a decrease in strength would suggest a reduction in protein function or stability.

Upon overexpression of wt-Vang, we observed consistent hair reorientation in specific regions of the wing. We therefore chose two regions that would be suitable for qualitative and quantitative analyses of the different mutants, with region 1 showing a more severe PCP phenotype as compared to region 2 (*Figure 2A*). Different mutations displayed different patterns of hair reorientation within these regions, confirming the system had an appropriate degree of sensitivity for this type of analysis. In region 1, D317E, R321L and K577H showed a distinct phenotype as compared to wt-Vang (over)expression. In fact, their hair pattern was more similar to that of *Vang*[-/-] wings or a genetically wild-type wing (*w*[1118]), included as a reference control that has no phenotype. In contrast, R332H, V391T and K418C all showed hair reorientation patterns similar to the gain-of-function (GOF) effect of wt-Vang overexpression (*Figure 2C* and *Figure 2—figure supplement 1B*). We also observed comparable changes in hair reorientation within region 2. In this case, D317E and K577H mirrored the phenotype of *Vang*[-/-], while R321L showed similarity to the reference control (*w*[1118]). R332H, V391T and K418C again showed similar patterns to the GOF overexpression of wt-Vang (*Figure 2D* and *Figure 2—figure supplement 1B*).

To quantify hair patterns, we utilized FijiWingsPolarity (*Dobens et al., 2018*), which determines the orientation of each hair within a given region, relative to the proximal-distal (P-D) axis. For visualization purposes, hair orientation angles are represented through arrows and a color gradient (*Figure 2C and D*). Through quantification of hair orientation angles, we were able to show that in region 1 all mutations with the exception of V391T had a significant alteration in hair reorientation as

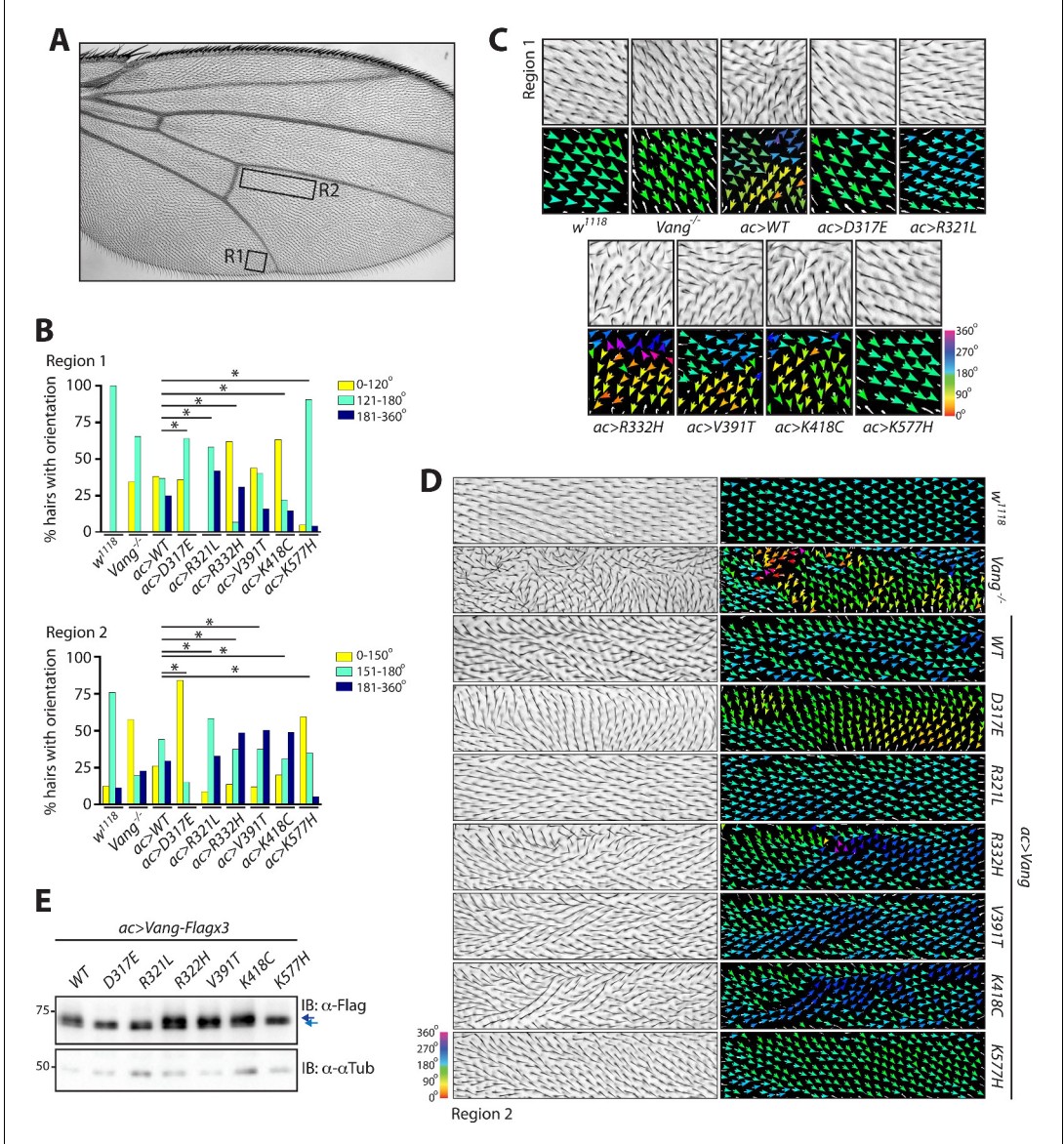

**Figure 2.** Behavior of over-expressed *Drosophila* Vang carrying C-terminal mutations. (A) Overview of an adult wing highlighting the analyzed regions of hair reorientation. (B) Quantification of the percentage of actin hairs oriented within a specific angle range upon *actin-Gal4* driven overexpression (*ac>*) of the different Vang mutant proteins and wild-type Vang. Also included for comparison is *Vang-/-* and *w1118* (a genetically wild-type wing that displays no phenotype). Graph is shown for region 1 (upper) and region 2 (lower). Angles from three independent wings were combined, and data binned to allow analysis using a *Chi*-squared test. * indicates p<0.001 (C) Example images of hair reorientation in region 1 in the indicated genotypes. Top panel shows adult wing, bottom panel shows the corresponding angles of hair orientation visualized through color (see 360 degree color scale, bottom right). Angles were determined using the Fuji plugin FijiWingsPolarity (*Dobens et al., 2018*). (D) Example images of hair reorientation in region 2. Left panel shows adult wing, and right panel the corresponding angles of hair orientation. (E) Western blot of wing disc lysates from the indicated genotypes. While similar levels of expression are observed, a difference in Vang mobility is notable for D317E, R321L and K577H, as indicated by blue arrows.

The online version of this article includes the following source data and figure supplement(s) for figure 2:

**Source data 1.** Raw data from the quantification of hair reorientation summarized in *Figure 2*.

**Figure supplement 1.** Expression of Vang transgenes mimics features of Vang gain- and loss-of-function phenotypes.

compared to wt-Vang overexpression, while in region 2 all mutations showed significant divergence from the wt-Vang (*Figure 2B*). To allow for statistical analysis, the data were pooled into three bins of angle orientation; normal orientation as determined by the reference control $w^{1118}$ (green), as well as a more anterior reorientation (yellow) or posterior reorientation (blue). The full spectrum of angles in 30° segments also reflected the changes in pattern (*Figure 2—figure supplement 1A*).

We confirmed that similar levels of protein expression were observed for each transgene (due to our experimental design). However, interestingly, D317E, R321L and K577H all showed a notable change in protein mobility (*Figure 2E*). This mobility shift was also observable in samples from *Vang-/-* tissue and in transfection experiments in S2 cells (*Figure 2—figure supplement 1C* and Figure 4C). It has previously been demonstrated that Vang is phosphorylated leading to a mobility change and associated band shift on gels (*Kelly et al., 2016*; *Strutt et al., 2019*). Accordingly, phosphatase treatment altered wt-Vang mobility, but it did not impact any of the mutants suggesting that their phosphorylation is reduced or lost (*Figure 2—figure supplement 1C*). As Vang phosphorylation is associated with membrane localization and function, this suggests that the D317E, R321L and K577H mutants are defective in one or both of these regards (*Kelly et al., 2016*; *Strutt et al., 2019*).

## Dominant behavior and protein localization defects in the majority of mutations

To further investigate the effect of the respective mutation in the overexpression system, we analyzed protein localization in the pupal wing at ~25 hr APF, by which time wt-Vang is predominantly localized to the plasma membrane (*Figure 3B*). Besides comparing the localization of the mutants themselves, we also assessed any dominant effects by examining their effect on the localization of co-expressed wt-Vang. We expressed wt-Vang directly via the *actin* promoter (ac-Vang-GFP) within physiological levels throughout the whole wing blade/animal and simultaneously used *engrailed(en)-Gal4* to overexpress the transgenes in the posterior compartment of the wing (*Figure 3A* and *Figure 3—figure supplement 1A*). This allowed for comparison of the localization of wt-Vang-GFP in the anterior (control) vs. posterior (transgene expression) compartments.

Overexpression of wt-Vang in the posterior compartment showed distinct membrane localization (as expected), visualized through Flag staining, while no signal was observed in the anterior compartment (*Figure 3B*). Vang-GFP (expressed from *ac-Vang-GFP*) also showed distinct membrane staining with levels reduced in the posterior compartment as compared to the anterior, suggestive of competition from the overexpressed Vang-Flag for membrane recruitment (*Figure 3B*). The junctional marker PatJ was unaffected by Vang expression (*Figure 3B*).

Consistent with their overexpression phenotypes, a number of changes to localization were observed for the different mutants. D317E and K577H were largely localized to the cytoplasm, and strikingly also altered the localization of wt-Vang-GFP in the posterior compartment (*Figure 3C*). In contrast, R321L behaved similarly to wt-Vang in both its localization and effect on Vang-GFP. R332H and K418C displayed a less distinct membrane localization that was echoed by diffuse Vang-GFP localization in the posterior compartment (*Figure 3C*). V391T showed distinct membrane localization with slightly increased membrane levels of Vang-GFP, as compared to controls (*Figure 3B and C*). In all cases, Vang-GFP and PatJ staining were unaffected in the anterior compartment (*Figure 3—figure supplement 1B*).

Overall, the localization data is consistent with our observed hair reorientation phenotypes, as well as protein mobility data (*Figure 2*). D317E and K577H behave like dominant negative, antimorph alleles. They cause hair reorientation patterns similar to *Vang^{-/-}* (*Figure 3D* and *Figure 3—figure supplement 1B*). The similarity in pattern to *Vang^{-/-}* can be explained, as in these like in *Vang^{-/-}* the protein is lacking from the membrane (*Figure 3C*). However, these mutants do not represent 'simple' LOF alleles, as they are dominantly affecting the localization of wt-Vang. R321L also displayed changes to protein mobility but did not lead to significant hair reorientation when overexpressed; its largely normal membrane localization and lack of effect on wt-Vang-GFP suggest it is a hypomorphic allele with reduced protein function (*Figures 2* and *3C*). R332H and K418C also appear to act in a dominant fashion, but in this case the protein is recruited to the membrane, where it could have a dominant effect on signaling (*Figure 3C*). In contrast, the staining of V391T and slight increase in accompanying Vang-GFP membrane localization suggest it may act as a mild GOF to enhance Vang signaling (*Figure 3C*).

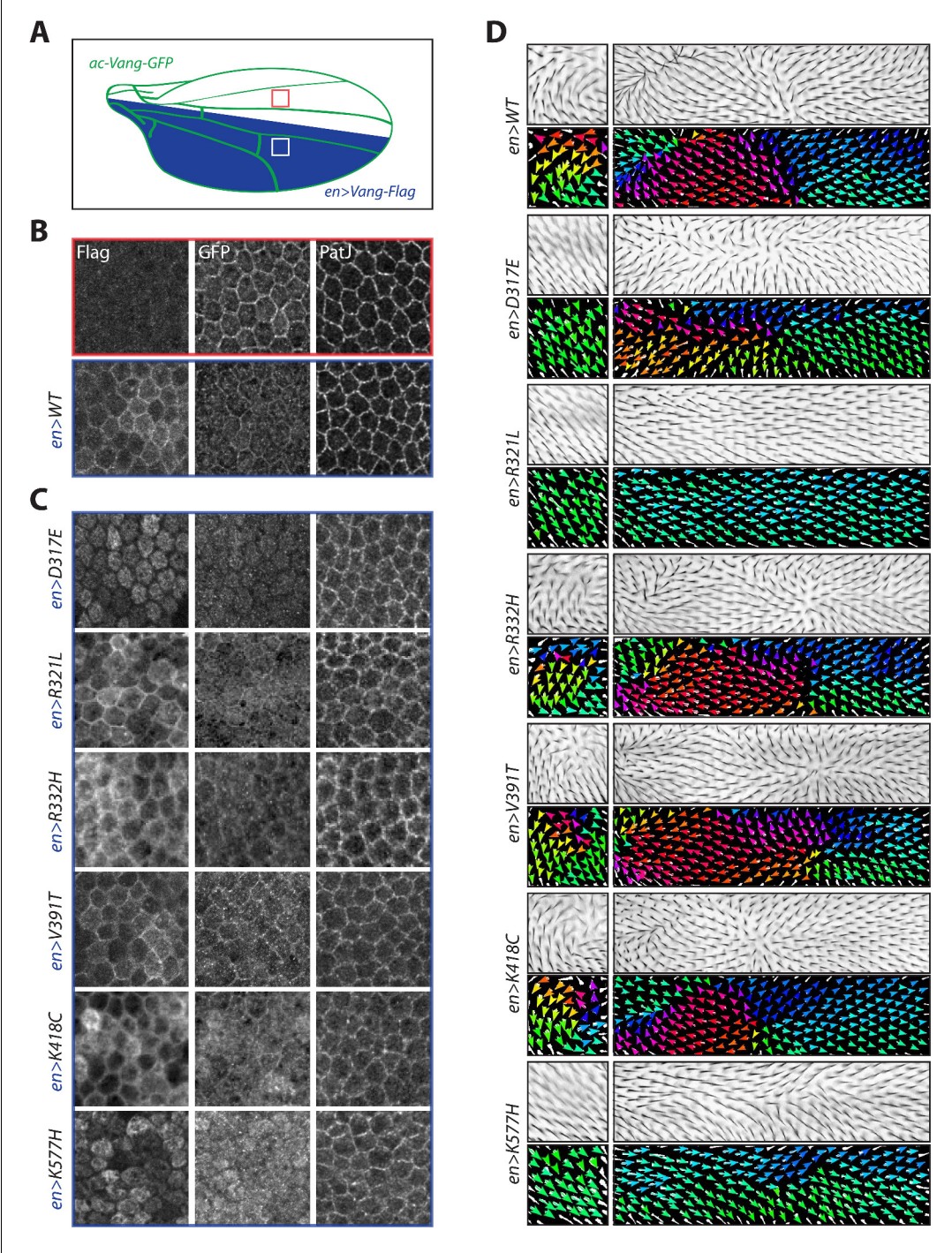

**Figure 3.** Overexpression of Vang transgenes affects localization of wild-type Vang. (A) Schematic showing regions of expression of different transgenes along the *Drosophila* wing blade. A direct *actin*-promoter driven (*ac-Vang-GFP*) construct is expressed at homeostatic levels throughout the wing blade, *engrailed-Gal4* drives expression of wild-type Vang-Flag or each of the associated mutant proteins in the posterior compartment of the wing. (B) Representative immunofluorescence images of regions from the anterior and posterior regions of a pupal wing overexpressing wild-type Vang-Flag in the posterior region. Note recruitment of Vang-Flag to the membrane in the posterior region (blue), and concomitant reduction in membrane localization of Vang-GFP in the posterior (blue) vs. anterior (red) compartment. The junctional marker PatJ shows consistent membrane labeling in both regions. (C) Representative immunofluorescence images of regions from the posterior compartment of pupal wings overexpressing the indicated Vang-Flag constructs. Note differences in localization of Vang-Flag mutants as compared to wild-type Vang (B), as well as altered wt-Vang-GFP localization. The junctional marker PatJ was unaffected in each case. (D) Regions of adult wing showing hair reorientation phenotypes in the indicated genotypes. Note the similarity in pattern between overexpression of WT, R332H, V391T and K418C mutants (swirls, cf. GOF in **Figure 2—**

*Figure 3 continued on next page*

*Figure 3 continued*

figure supplement 1), as compared to R321L (wild-type orientation), and D317E and K577H (downwards reorientation, cf. LOF in *Figure 2—figure supplement 1*) in posterior region of the wing. Color range is same as in *Figure 2*.

The online version of this article includes the following figure supplement(s) for figure 3:

**Figure supplement 1.** Expression of *ac-Vang-GFP* within the *Drosophila* wing blade does not interfere with polarity.

## Mutants show altered membrane localization in rescue conditions

To confirm that the localization of the mutant proteins was not due to interference from endogenous Vang, we next assayed its localization in a *Vang*[-/-] background. Here, we used *actin*-Gal4 driven expression with which, due to the higher than endogenous expression levels of the transgenes (via the Gal4 amplification), we observed a gain-of-function phenotype with wt-Vang (*Figure 4A*). Despite this effect, we observed hair reorientation patterns in the absence of endogenous Vang that were consistent with the experiments described above. D317E and K577H displayed hair reorientation patterns similar to *Vang*[-/-], while R332H, V391T and K418C showed a similar phenotype of hair orientation swirls as compared to overexpressed wt-Vang. Furthermore, R321L showed a more 'complete rescue', supporting its behavior as a hypomorph in protein function and thus a 'weaker GOF' phenotype, which resembles a better rescue of the *Vang*[-/-] background (*Figure 4A* and *Figure 4—figure supplement 1B*).

Analysis of Flag staining for the transgenes revealed similar localization patterns as compared to the overexpression experiment in a wt control background (*Figure 3*). Wt-Vang localized distinctly at the membrane, indicated by an overlap with cortical actin stain (*Figure 4B*), while R321L and V391T showed slightly reduced and enhanced membrane localization, respectively. Furthermore, R332H and K418C displayed a diffuse membrane localization, and D317E and K577H failed to localize to the membrane (*Figure 4B*). We again confirmed that expression levels of each transgene were equivalent, and expected changes in protein mobility were detectable (*Figure 4C*).

Taken together, our data are consistent with the conclusions that D317E and K577H act as dominant-negative mutations with reduced membrane localization, while R332H and K418C act as dominant mutations that exert their effect at the membrane. R321L behaves like a hypomorph with reduced protein function, and V391T is a mild GOF enhancing protein function.

## Rescue experiments in the eye confirm loss-of-function and gain-of-function behavior

To confirm and further refine the suggested mechanistic effects of mutations in altering Vang function, we analyzed their phenotype in the absence of endogenous Vang in an additional tissue. For this, we used the *Drosophila* eye, which shows generally weaker GOF effects with high expression levels of Vang as compared to the wing.

For consistency, we used *actin-Gal4* (*ac >Vang*) to express the transgenes, and we also performed experiments with *sep-Gal4*, an eye specific driver related to and based on the *sevenless/sev* enhancer and promoter, that gives lower levels and spatially restricted expression (*Fanto et al., 2000*). PCP phenotypes in the eye are characterized by changes in ommatidial chirality as well as orientation (*Figure 5A*). Introduction of *ac >Vang* Flag in the *Vang*[-/-] background resulted in a near perfect rescue with very minor chirality and orientation defects. As this phenotype was trending toward an overexpression phenotype of Vang, this suggested that the levels were just marginally too high for a complete rescue (*Figure 5B*). In contrast, due to its weaker expression *sep >Vang* Flag showed only a partial rescue (*Figure 5B*). By comparing phenotypes with both Gal4 drivers, we were thus able to better ascertain how the mutants affected Vang function.

With both drivers, D317E and K577H showed very limited rescue in regards to chirality and only partial rescue in orientation, overall displaying a similar phenotype to *Vang*[-/-] (*Figure 5C,D* and *Figure 5—figure supplement 1*). This is consistent with their antimorphic nature; however, the partial rescue we observe suggests the mutants retain some activity. With *actin-Gal4*, R321L displayed a significantly better rescue of chirality (as compared to wt-Vang control), while with *sep-Gal4* the rescue was less efficient both in terms of chirality and orientation (*Figure 5C,D* and *Figure 5—figure supplement 1*). Together this confirmed that R321L is a mild hypomorphic LOF allele and that the better rescue level observed with *ac-Gal4*, as compared to wt-Vang, is due to the higher expression

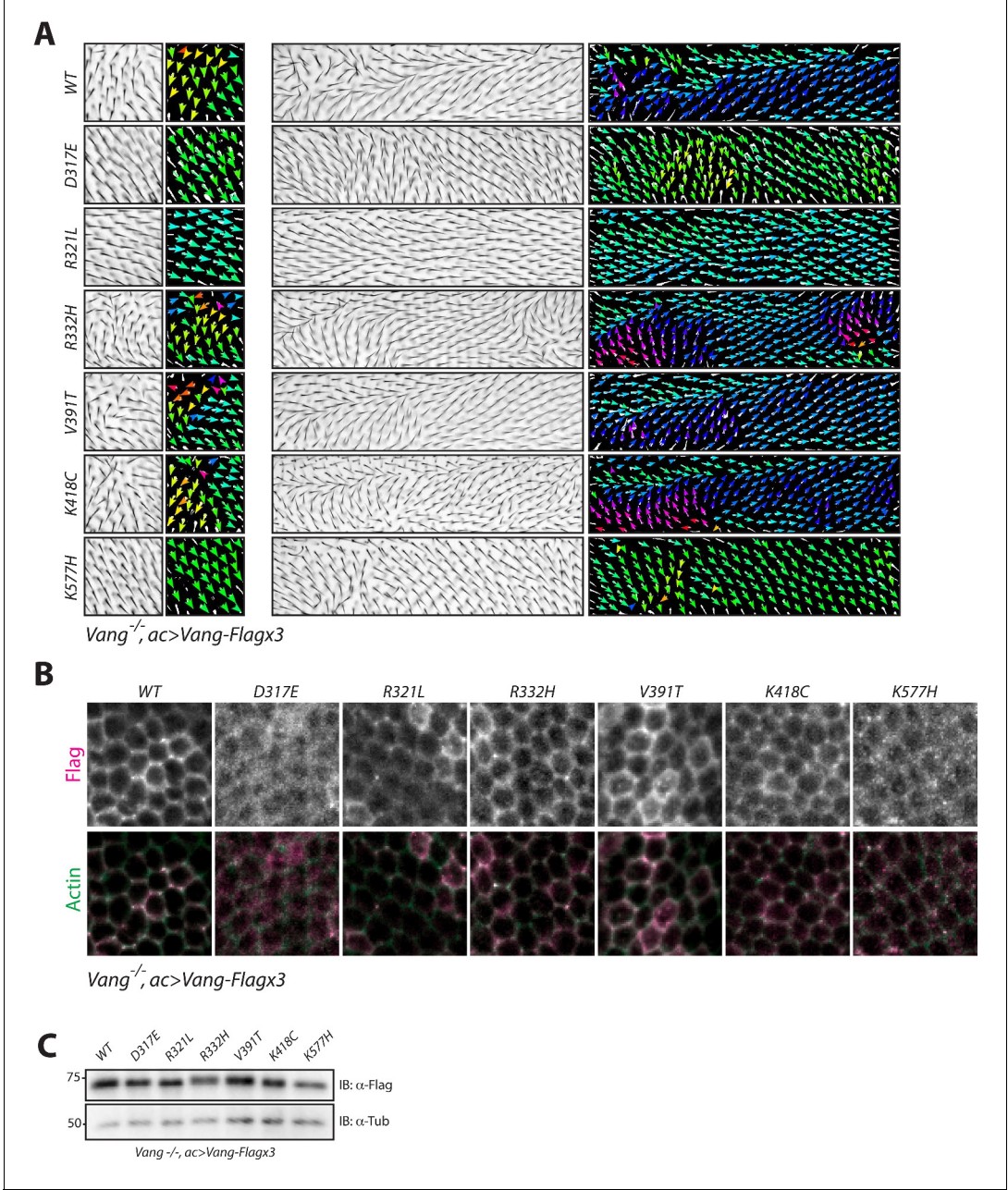

**Figure 4.** NTD mutations affect membrane localization of Vang. (**A**) Regions 1 and 2 of the adult wing (see *Figure 2A*) showing hair reorientation phenotypes in the indicated genotype. Note similarity in pattern between WT, R332H, V391T and K418C mutants (swirls, cf. GOF in *Figure 2—figure supplement 1*), as compared to R321L (wild-type orientation), and D317E and K577H (downwards reorientation, cf. LOF in *Figure 2—figure supplement 1*). Color range is same as in *Figure 2*. (**B**) Representative immunofluorescence images of pupal wings upon expression of Vang-Flag or indicated mutant Vang proteins using *actin-Gal4* driver. Note changes to Vang localization as shown through Flag staining (red), and the degree of overlap with actin (stained with Phalloidin, green) which marks the membrane (**C**) Western blot of wing discs from the indicated genotypes. Note that all transgenes were expressed at similar levels. Differences in mobility are also observed as in *Figure 2* and *Figure 2—figure supplement 1*. The online version of this article includes the following figure supplement(s) for figure 4:

**Figure supplement 1.** Schematic representation of wings shows flow of hair reorientation in different mutant conditions.

levels. R332H showed a significant enhancement in chiral defects, as compared to control wt-Vang, with the *actin*-driver, as well as more significant orientation defects with the *sep*-driver. K418C showed a phenotype similar to wt-Vang (*Figure 5C,D* and *Figure 5—figure supplement 1*). While, V391T trended towards more severe chiral defects with the *actin*-driver, with a significantly better

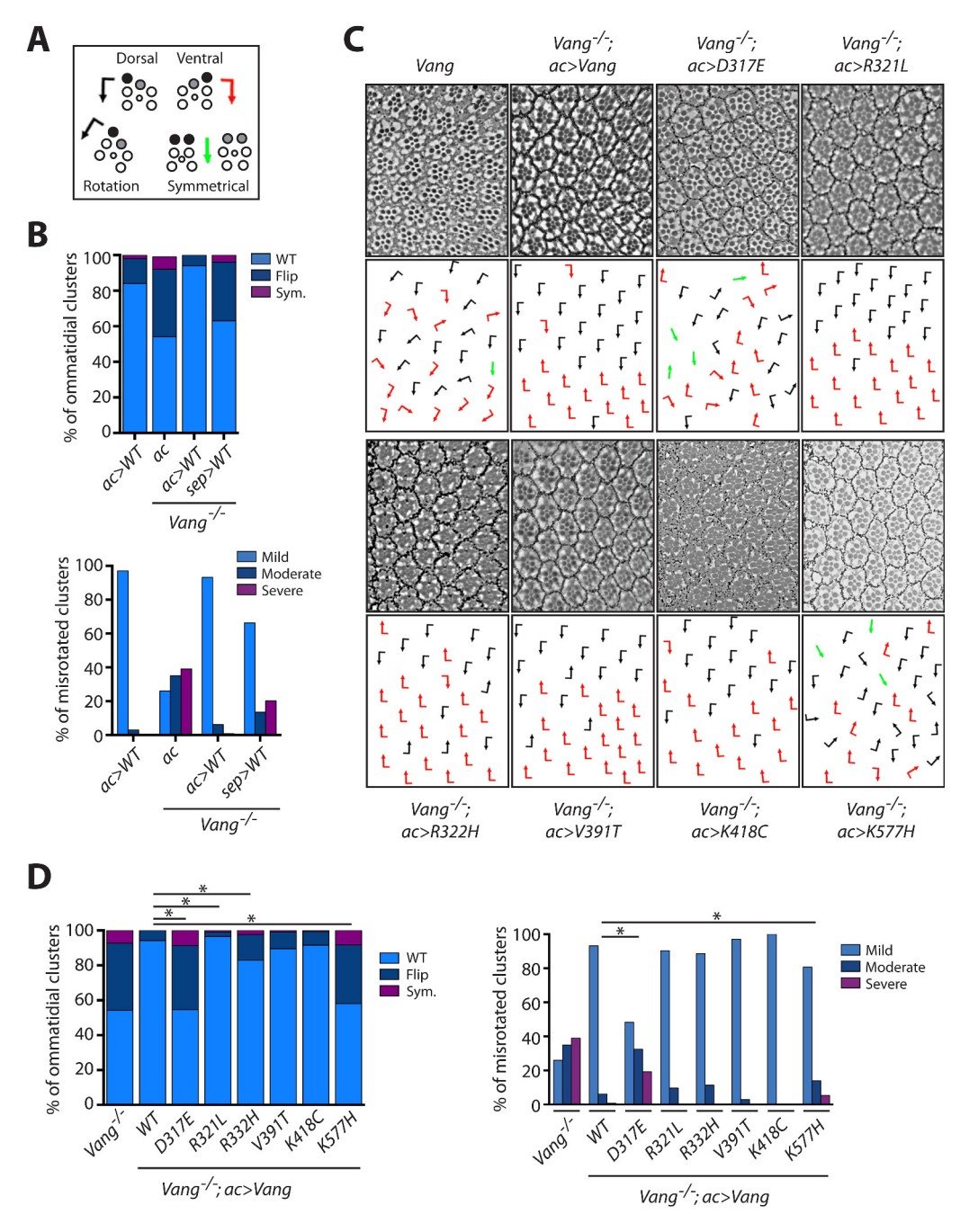

**Figure 5.** *Vang-/-* rescue experiments with individual mutations reveal the nature of how they affect Vang function. (**A**) Schematic representation of the chiral forms of ommatidial clusters within the *Drosophila* eye. Dorsal chirality is represented by black arrows, ventral chirality by red arrows, and symmetrical (achiral) clusters by green arrows. The varying degree to which each cluster rotated during development is denoted by the angle of each arrow. (**B**) Quantification of the percentage of different chiral forms of clusters in the indicated genotypes (top). These include wild-type (WT), a flipped chiral form (Flip), or a symmetrical cluster (Sym.). Quantification of the percentage of misrotated clusters in the associated genotypes (bottom). Misrotation was split into three categories, mild is wild-type +/- 10 degrees, moderate is +/- 10–30 degrees from wild-type, and severe is +/- > 30 degrees misrotation from wild-type. Note that rescue with *actin-Gal4* results in a mild overexpression phenotype due to the expression strength with this driver, (also ***Figure 5—figure supplement 1***), while *sep-Gal4* shows a weaker rescue due to its weaker and restricted (not all R-cells) expression. (**C**) Tangential eye sections of the region flanking the dorso-ventral midline (equator) in the indicated genotypes after rescue with *actin-Gal4* driver, anterior is left and dorsal is up. (**D**) Quantification of percentage of different chiral forms of clusters in the indicated genotypes (left). Quantification of percentage of misrotated clusters in the indicated genotypes (right). Data were analyzed using a *Chi*-squared test: * indicates p<0.001.

*Figure 5 continued on next page*

*Figure 5 continued*

The online version of this article includes the following source data and figure supplement(s) for figure 5:

**Source data 1.** Raw data from the quantification of eye chirality and rotation defects summarized in *Figure 5*.

**Figure supplement 1.** Phenotypes observed upon partial rescue with sep-Gal4 are consistent with results obtained using actin-Gal4.

rescue of orientation with the *sep*-driver (*Figure 5C,D* and *Figure 5—figure supplement 1*). Taken together with the wing data, this supports the notion that V391T is a mild hypermorph with GOF activity.

## Mutations associated with reduced membrane localization or loss in protein function disrupt effector binding

Due to the defects associated with the different mutants, we wished to determine how they might affect effector binding. As many of the mammalian studies have focused on the interaction between Dishevelled (Dvl) and Vangl, we first examined the binding between Dsh and Vang, the *Drosophila* orthologues. We expressed wt-Vang-Flag and mutant constructs in S2 cells and tested for their interaction with Dsh-GFP by pull-down. D317E, R321L and K577H all showed a reduced ability to interact with Dsh-GFP (*Figure 6A*), while normal binding, comparable to wt-Vang, was observed for R332H, V391T and K418C (*Figure 6B*). Next we tested binding between the different Vang mutants and Prickle (Pk), the main effector and cytoplasmic interaction partner of Vang in vivo. The same pattern of binding was observed, with a reduction in binding to D317E, R321L and K577H, but no change with R332H, V391T and K418C (*Figure 6C and D*). This is consistent with the affected membrane association of the first three Vang mutations and our previous functional and phenotypic assertions (summarized in *Figure 6E*).

Furthermore, these interaction data suggest a trend whereby D317E, R321L and K577H all lead to a general reduction in effector binding. Interestingly, analyzing the location of these mutants along the C-terminal tail, they map in close proximity to regions of specific effector binding sites (*Figure 6G*). K557H is situated near to the PDZ binding motif which is involved in Scribble (Scrib) binding (Scrib is required for PCP and has been shown to interact with Vang in both mice and *Drosophila* [*Courbard et al., 2009*; *Montcouquiol et al., 2003*]). We were able to show that Dgo binding requires amino acids 304–323 (*Figure 6F*) and D317E and R321L map within this region. As demonstrated, these mutations do not show specificity for a particular effector but instead a general reduction in binding. This correlation suggests that sites of effector binding along the C-terminal tail highlight regions of high sensitivity for Vang functionality, not just for binding to a particular effector, but in general for the integrity of the entire protein.

## Discussion

Our analyses revealed a differential and causative behavior for the NTD-associated Vangl mutants. *Drosophila* provides unprecedented depth to the analysis of PCP in the respective mutations through its well-established and detailed in vivo PCP features, the possibilities to study the NTD-mutations in an otherwise wild-type background as well as in the *Vang-/-* context, and both quantitative and qualitative nature of the assays for PCP. Taken together, our data revealed that the mutations analyzed fall into different functional categories and that all patient derived mutations tested are indeed causative of PCP defects, and hence likely causative of NTD in the patients.

### Functional definition of both dominant negative and gain-of-function mutations

The detailed analyses of the NTD-associated mutations revealed specific functional features. For example, the R321L mutation (all numbers refer to the residues in *Drosophila* Vang, see *Figure 1* for the respective numbers in mouse and human, unless otherwise indicated) behaves like a mild hypomorphic LOF allele, as demonstrated largely by a lack of phenotype upon over-expression and through the level of rescue observed. This is consistent with data from the *Vangl2^{LP}*-R259L mouse, where only mild NTD defects were observed, along with no difference in membrane localization or levels (*Guyot et al., 2011*). In our system, besides reduced membrane localization, we also observed

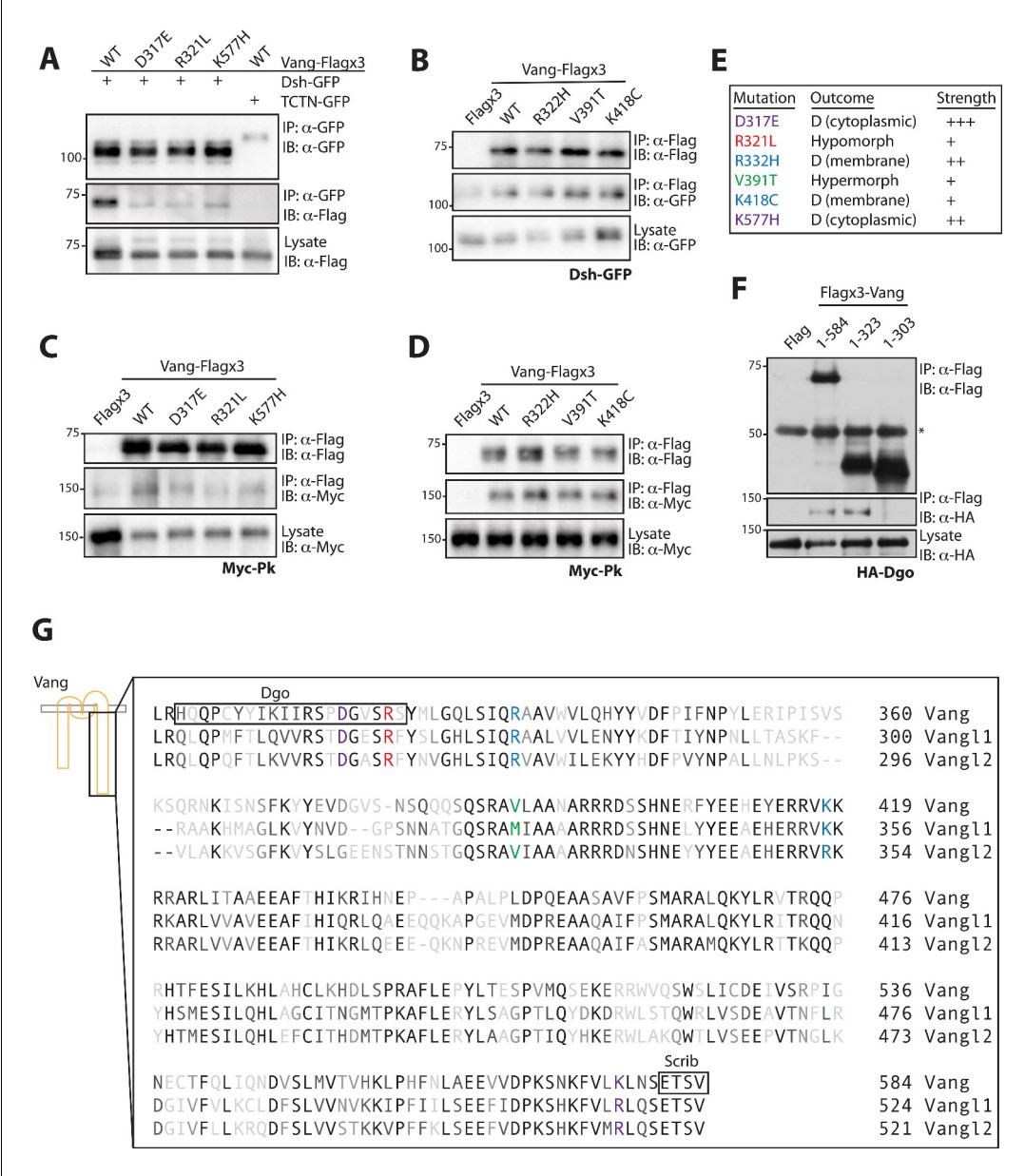

**Figure 6.** Mutations associated with LOF phenotypes display reduced effector binding. (**A**) Western blot showing binding between Dsh-GFP and Vang-Flag wild-type or the indicated mutants. Note that Vang-Flag wild-type binds to Dsh-GFP but not the negative control Tctn-GFP. The D317E, R321L and K557H mutants show reduced binding as compared to wild-type Vang. (**B**) Western blot showing binding between Vang-Flag wild-type or the indicated mutants and Dsh-GFP. The Vang constructs are able to bind to Dsh to the same degree, while no binding is observed with Flag alone. (**C**) Western blot showing binding between Vang-Flag wild-type or the indicated mutants and Myc-Pk. The D317E, R321L and K577H mutants show a markedly weaker interaction with Myc-Pk as compared to wild-type Vang. (**D**) Western blot showing binding between Vang-Flag wild-type or the indicated mutants and Myc-Pk. The Vang constructs are able to bind Pk to the same degree. (**E**) Summary of the phenotypic outcome of investigated mutations, D, (cytoplasmic) and (membrane) distinguish between the two types of dominant mutations observed, which is based on their cellular localization. Colors are assigned based on different outcomes, also shown is the relative phenotypic strength of each mutation (see **Figures 3** and **4**). (**F**) Western blot showing the regions of Vang required for interaction with Dgo. Note that full-length Vang (1-584) and a construct containing residues 1–323 of Vang are able to interact with HA-Dgo. While a construct containing residues 1–303 does not interact, suggesting residues 304–323 in Vang are essential for its interaction with Dgo. (**G**) Schematic showing the C-terminal sequence of Vang, and human Vangl1 and Vangl2. Mutated residues are highlighted in colors associated with their phenotypic outcome as in E. Also highlighted are regions of Vang essential for its interaction with Dgo and Scribble, note their proximity to specific residues.

its aberrant mobility in western blot assays, an indication that functionality of the protein was decreased, as well as diminished ability to bind effectors. Differences in protein mobility, localization, and effector binding were also observed for D317E and K577H. In contrast to R321L, however, both mutants failed to localize to the membrane, which was reflected in a hair orientation phenotype reminiscent of a *Vang*[-/-] wing. Furthermore, both mutants showed largely no rescue in the respective assays. D317E displayed a stronger phenotype as compared to K577H, which correlated with the severity of subcellular mislocalization, highlighting the sensitivity and accuracy of the *Drosophila* assays.

Expression of D317E and K577H transgenes also led to mislocalization of wild-type Vang, revealing mechanistic insight into the dominant nature of these mutations. For D317E, this is consistent with data from the mouse (D255E), with our data further demonstrating that the mutation does indeed act in a dominant negative manner. It was previously demonstrated that Sec24b promotes the selective sorting of Vangl2 into vesicles for ER to Golgi transport, and that the *Vangl2*^*Lp(D255E)*^ allele fails to undergo this process (*Merte et al., 2010*). It has been demonstrated that the *Vangl2*^*LP*^ mouse shows reduced membrane protein levels, thought to result from the protein becoming trapped in the ER and subsequently targeted for degradation. While this might suggest a 'simple' LOF phenotype, the ability of Vangl1 and Vangl2 to interact complicates matters (*Yin et al., 2012*). Further, it was demonstrated that Vangl1 is missing from the membrane in *Vangl2*^*LP*^ mice but not in a *Vangl2* knock-out, with *Vangl2*^*LP*^ also displaying a more severe phenotype (*Song et al., 2010*; *Yin et al., 2012*). Additionally, a dominant effect was observed in regard to the phosphorylation of wild-type Vangl2, which was reduced upon co-expression of the *Vangl2*^*LP*^ mutant (*Gao et al., 2011*). This is consistent with - and reminiscent of - our dominant relocalization observations and, due to the hair reorientation pattern caused in our experiments, our data support the rationale that the original *Vangl2*^*LP*^ alleles are dominant negative mutations and not LOF. This is in alignment with the original genetic studies performed in the mouse that proposed semi-dominance (*Copp et al., 1994*; *Strong and Hollander, 1949*).

The patient derived K577H mutation had not previously been investigated. Here, we demonstrate that it is causative for PCP defects, and interestingly showed mechanistic similarity to D317E (the original *Vangl2*^*LP*^ mutation, see above), functioning like a dominant negative. Therefore, our data provide a link between the NTD mouse mutations and a human mutation and support the idea that mutations that affect PCP signaling in distinct ways could all contribute to NTD pathology. In line with this notion, we also observed that R332H and K418C displayed dominant activity in functional assays, consistent with the hypothesis that they are also causative in NTD. However, their phenotypic impact was different. These mutants retained effector binding and were localized to the membrane, even if not as distinctly as wild-type Vang. Additionally, the hair re-orientation phenotype observed was reminiscent of a GOF effect of Vang, which results from the non-polar localization of core PCP proteins at the membrane. As their interaction with cytoplasmic effectors and phosphorylation appear to be retained, this points to R332H and K418C behaving as hypermorphs with an effect on the PCP signaling complexes at the membrane. However, due to the relatively high expression levels of our transgenes we were not able to find conditions to analyze changes to protein asymmetry. Nonetheless, taken together and most importantly, the above-mentioned human patient mutations, cause dominant PCP defects, K577H with a dominant negative function, and R332H and K418C as hypermorphic alleles, and thus are likely causative of the NTDs.

Finally, we obtained data suggestive of V391T functioning as a mild hypermorph GOF. Although in the majority of assays V391T behaved similarly to R332H and K418C, its membrane localization appeared more robust, and rescue experiments were suggestive of the mutation leading to a milder GOF, and not dominant behavior, as was seen with R332H and K418C. Nonetheless, the data for this human mutation is again consistent with it also being causative of NTD.

## *Drosophila* as model to define causative nature of human mutations

While there are caveats of using any specific system, including *Drosophila*, the ability to phenocopy and refine results suggested from the mouse, for example in the case of the original *Vangl2*^*LP*^, and to demonstrate sensitivity through quantitative analyses, highlights its utility in exploring the functionality and potential causative nature of human NTD-associated mutations. In fact, our assays were both able to reveal additional functional insight into mutations that had been studied previously by other means and to define functional and mechanistic behavior of previously unstudied human

mutations. A popular system to investigate the different mutations is to analyze their localization in MDCK cells. In this assay, the mouse *Vangl2^LP* allele, D255E, was detected within the ER, and showed reduced stability and proteosomal degradation, echoing phenotypes from the mouse (*Gravel et al., 2010*). The same assay also showed defective localization for S464N, R259L and R274Q (equivalent to R332H in our assay), with all mutations displaying very similar phenotypes (*Iliescu et al., 2014*; *Iliescu et al., 2011*). While a useful assay to reveal potentially damaging effects upon *Vangl1/2*, it does not capture the true nature of the mutation as our in vivo assays do. For example, R259L is milder in phenotype as compared to S464N and displays membrane localization in vivo, which is not captured in the MDCK assay. In contrast, our assays capture these features in the *Drosophila* in vivo system. Furthermore our results define that R274Q/R270H in *hVANGL1/2*, respectively (R332H in *Drosophila*), likely has dominant negative activity. The result in MDCK cells for both R259L and R274Q is also at odds with experiments performed in zebrafish. Both mutants behaved similarly to wild-type protein in overexpression and rescue experiments (*Guyot et al., 2011*; *Reynolds et al., 2010*). The lack of phenotype from the zebrafish experiments suggests that, while suitable for mutations that have a strong pathological effect, this system may not be sensitive enough to reveal phenotypic and functional insight with milder mutations.

In line with this hypothesis, the zebrafish system did show phenotypic behavior for the *Vangl2^LP* mutations, mouse D255E and S464N, and additionally human M328T, with all performing as LOF in assays in the fish (*Reynolds et al., 2010*). Our data also revealed that mouse D255E (D317E in *Drosophila*) displayed LOF characteristics, for example diminished effector binding. However, our assays allow us to conclude further, as discussed above, that the mutant displays dominant negative function. In contrast, our data are not consistent with the zebrafish findings for human M328T. The equivalent *Drosophila* mutant, V391T, displayed a mild GOF phenotype. This residue was the least conserved among the mutations tested, and while it is methionine in human and mouse Vangl1, the equivalent residue is valine in mouse, human and frog Vangl2, as well as in *Drosophila* Vang. The differences in phenotypic outcome may be due to this discrepancy, and thus investigating the phenotypic behavior of Vangl2 with an equivalent mutation in a mammalian assay could provide further insight. This result also implies, as is to be expected, that the most conserved residues between systems will give the most reliable functional insight. Nonetheless, we were able to define that a mutation at this residue causes a dominant PCP defect, suggesting its importance for overall Vang function in PCP in general and NTD in particular.

As discussed above, our data set reveals graded phenotypes in vivo, with different mutations displaying more or less severe PCP defects. However, we could not find a pattern to correlate the severity observed with features of the different residues, for example, whether they occurred originally in *VANGL1* vs. *VANGL2*, if the substitution was more conservative or drastic, or whether familial or sporadic. This suggests that it is the location of the residue that exhibits a mutational change as the most important factor in determining phenotypic strength. A prime example of this is the comparison of the two mouse *Vangl2^LP* alleles with a conservative substitution D255E and more drastic substitution R259L. While they are located in a similar region, the closer proximity of D255E downstream of the transmembrane domain, which also correlates with the Dgo interaction region, suggests it is a functionally more important residue and site, for protein integrity. Consistent with this notion, our observations that the mutations that display LOF characteristics, D255E, R259L and K577H, all map close to regions of effector binding, suggest that these regions are important for the general functional and structural integrity of Vang family proteins. While we cannot rule out that the reduced binding observed for D317E and K577H is due to their mislocalization and inaccessibility to effector proteins, this is clearly not the case for R321L which is localized to the membrane. We therefore favor the above hypothesis and believe these mutants lead to a general structural defect. While many of the NTD mutations may interfere with Vang/Vangl function by affecting its structure, it is also possible that they hit as yet unknown motifs or sites of post-translational modification. For example, R274/270, in human *VANGL1/2*, respectively (R332 in *Drosophila*), forms part of a putative D-box ubiquitination motif (*Iliescu et al., 2014*), and two mutations found in the N-terminus of human patients are mapped to phosphorylation sites important for Vang function (*Gao et al., 2011*; *Kibar et al., 2009*; *Lei et al., 2010*). Overall, this suggests that there is much more to be revealed in determining both the contribution of PCP mutants to NTD, and that analyzing such NTD-associated mutations will reveal important mechanistic insight into Vang/Vangl function in general, in both developmental and disease contexts.

# Materials and methods

## Key resources table

| Reagent type (species) or resource | Designation | Source or reference | Identifiers | Additional information |
|---|---|---|---|---|
| Gene (*Drosophila melanogaster*) | *Vang* | Pubmed ID: 35922 | Dmel_CG807, CG8075, Dmel\CG8075, Stbm, Strabismus, Van Gogh | Chromosome 2R, NT_033778.4 (9103238...9106796) |
| Genetic reagent (*Drosophila melanogaster*) | *w[1118]* | FlyBase ID: FBal0018186 | | Reference control |
| Genetic reagent (*Drosophila melanogaster*) | *Vang[6]* | Bloomington *Drosophila* Stock Center | 6918 | Null allele |
| Genetic reagent (*Drosophila melanogaster*) | *UAS-Vang- Flagx3* | This Paper | | Insertion into BDSC stock 9752 - PBAC{yellow[+]-attP-3B}VK00037 Can be obtained from Mlodzik Laboratory, ISMMS |
| Genetic reagent (*Drosophila melanogaster*) | *UAS-Vang-D317E-Flagx3* | This Paper | | Insertion into BDSC stock 9752 - PBAC {yellow[+]-attP-3B} VK00037 Can be obtained from Mlodzik Laboratory, ISMMS |
| Genetic reagent (*Drosophila melanogaster*) | *UAS-Vang-R321L-Flagx3* | This Paper | | Insertion into BDSC stock 9752 - PBAC{yellow[+]-attP-3B}VK00037 Can be obtained from Mlodzik Laboratory, ISMMS |
| Genetic reagent (*Drosophila melanogaster*) | *UAS-Vang-R332H-Flagx3* | This Paper | | Insertion into BDSC stock 9752 - PBAC{yellow[+]-attP-3B}VK00037 Can be obtained from Mlodzik Laboratory, ISMMS |
| Genetic reagent (*Drosophila melanogaster*) | *UAS-Vang-V391T-Flagx3* | This Paper | | Insertion into BDSC stock 9752 - PBAC{yellow[+]-attP-3B}VK00037 Can be obtained from Mlodzik Laboratory, ISMMS |
| Genetic reagent (*Drosophila melanogaster*) | *UAS-Vang-K418C-Flagx3* | This Paper | | Insertion into BDSC stock 9752 - PBAC{yellow[+]-attP-3B}VK00037 Can be obtained from Mlodzik Laboratory, ISMMS |
| Genetic reagent (*Drosophila melanogaster*) | *UAS-Vang-K577H-Flagx3* | This Paper | | Insertion into BDSC stock 9752 - PBAC{yellow[+]-attP-3B}VK00037 Can be obtained from Mlodzik Laboratory, ISMMS |
| Genetic reagent (*Drosophila melanogaster*) | *Vang[6],UAS-Vang-Flagx3* | This Paper | | Recombined stock Can be obtained from Mlodzik Laboratory, ISMMS |
| Genetic reagent (*Drosophila melanogaster*) | *Vang[6],UAS-Vang-D317E-Flagx3* | This Paper | | Recombined stock Can be obtained from Mlodzik Laboratory, ISMMS |
| Genetic reagent (*Drosophila melanogaster*) | *Vang[6],UAS-Vang-R321L-Flagx3* | This Paper | | Recombined stock Can be obtained from Mlodzik Laboratory, ISMMS |

*Continued on next page*

*Continued*

| Reagent type (species) or resource | Designation | Source or reference | Identifiers | Additional information |
|---|---|---|---|---|
| Genetic reagent (*Drosophila melanogaster*) | *Vang[6],UAS-Vang-R332H-Flagx3* | This Paper | | Recombined stock Can be obtained from Mlodzik Laboratory, ISMMS |
| Genetic reagent (*Drosophila melanogaster*) | *Vang[6],UAS-Vang-V391T-Flagx3* | This Paper | | Recombined stock Can be obtained from Mlodzik Laboratory, ISMMS |
| Genetic reagent (*Drosophila melanogaster*) | *Vang[6],UAS-Vang-K418C-Flagx3* | This Paper | | Recombined stock Can be obtained from Mlodzik Laboratory, ISMMS |
| Genetic reagent (*Drosophila melanogaster*) | *Vang[6],UAS-Vang-K577H-Flagx3* | This Paper | | Recombined stock Can be obtained from Mlodzik Laboratory, ISMMS |
| Genetic reagent (*Drosophila melanogaster*) | *actin-Gal4* | Bloomington *Drosophila* Stock Center | 3954 | |
| Genetic reagent (*Drosophila melanogaster*) | *ac-Vang-GFP* | Gift from David Strutt, University of Sheffield, UK | | |
| Genetic reagent (*Drosophila melanogaster*) | *en-Gal4* | Bloomington *Drosophila* Stock Center | 1973 | |
| Genetic reagent (*Drosophila melanogaster*) | *sep-Gal4* | (*Fanto et al., 2000*) | | |
| Cell line (*Drosophila melanogaster*) | S2 | Thermo Fisher Scientific | 69007 | Stock tested for contamination, characterized by isozyme and karyotype analysis |
| Transfected construct (*Drosophila melanogaster*) | pUAS-Vang-Flagx3 | This Paper (*Bischof et al., 2007*) | pUASTattB vector | Cloned using NotI-XbaI Can be obtained from Mlodzik Laboratory, ISMMS |
| Transfected construct (*Drosophila melanogaster*) | pUAS-Vang-D317E-Flagx3 | This Paper | | Made using SDM Can be obtained from Mlodzik Laboratory, ISMMS |
| Transfected construct (*Drosophila melanogaster*) | pUAS-Vang-R321L-Flagx3 | This Paper | | Made using SDM Can be obtained from Mlodzik Laboratory, ISMMS |
| Transfected construct (*Drosophila melanogaster*) | pUAS-Vang-R332H-Flagx3 | This Paper | | Made using SDM Can be obtained from Mlodzik Laboratory, ISMMS |
| Transfected construct (*Drosophila melanogaster*) | pUAS-Vang-V391T-Flagx3 | This Paper | | Made using SDM Can be obtained from Mlodzik Laboratory, ISMMS |
| Transfected construct (*Drosophila melanogaster*) | pUAS-Vang-K418C-Flagx3 | This Paper | | Made using SDM Can be obtained from Mlodzik Laboratory, ISMMS |

Continued

| Reagent type (species) or resource | Designation | Source or reference | Identifiers | Additional information |
|---|---|---|---|---|
| Transfected construct (*Drosophila melanogaster*) | pUAS-Vang-K577H-Flagx3 | This Paper | | Made using SDM Can be obtained from Mlodzik Laboratory, ISMMS |
| Transfected construct (*Drosophila melanogaster*) | pAc5.1-Gal4 | Gift from Andreas Jenny, AECOM, USA | | |
| Transfected construct (*Drosophila melanogaster*) | pAc5.1-Dsh-GFP | (*Simons et al., 2009*) | | |
| Transfected construct (*Drosophila melanogaster*) | pAc5.1-Myc-Pk | Gift from Andreas Jenny, AECOM, USA | | |
| Transfected construct (*Drosophila melanogaster*) | pAc5.1-HA-Dgo | Gift from Andreas Jenny, AECOM, USA | | |
| Transfected construct (*Drosophila melanogaster*) | pTub-Tctn-GFP | This Paper | pCaSpeRTubGFP vector with pUAST MCS | Cloned using BglII-XhoI Can be obtained from Mlodzik Laboratory, ISMMS |
| Transfected construct (*Drosophila melanogaster*) | pAc5.1-Flagx3 | Gift from Andreas Jenny, AECOM, USA | | |
| Transfected construct (*Drosophila melanogaster*) | pAc5.1-Flag-Vang | This Paper | pAc5.1-Flag vector | Cloned using NotI-XbaI Can be obtained from Mlodzik Laboratory, ISMMS |
| Transfected construct (*Drosophila melanogaster*) | pAc5.1-Flag-Vang 1–323 | This Paper | pAc5.1-Flag vector | Cloned using NotI-XbaI Can be obtained from Mlodzik Laboratory, ISMMS |
| Transfected construct (*Drosophila melanogaster*) | pAc5.1-Flag-Vang-1–303 | This Paper | pAc5.1-Flag vector | Cloned using NotI-XbaI Can be obtained from Mlodzik Laboratory, ISMMS |
| Antibody | Flag | Sigma Aldrich | M2 | 1:5000 IB/I:50 IF |
| Antibody | Gamma-Tubulin | Sigma Aldrich | GTU-88 | 1:1000 |
| Antibody | GFP | Roche | 7.1 and 13.1 | 1:1000 |
| Antibody | GFP | Invitrogen | A11122 | 1:100 |
| Antibody | PatJ | Gift from Jun Wu, ISMMS, USA | | 1:500 |
| Antibody | Myc | Santa Cruz Biotechnology | 9E10 | 1:1000 |
| Antibody | HA | Roche | 3F10 | 1:1000 |
| Sequence-based reagent | D317E | GATCATTCGCTCCCCG GAAGGCGTTTCGCGCTCCTAC | | PCR primer |
| Sequence-based reagent | R321L | GACGGCGTTTCGCT CTCCTACATGTTG | | PCR primer |
| Sequence-based reagent | R332H | GTCAGCTGAGCATCCAAC ATGCGGCTGTGTGGGTGCTAC | | PCR primer |

*Continued on next page*

*Continued*

| Reagent type (species) or resource | Designation | Source or reference | Identifiers | Additional information |
|---|---|---|---|---|
| Sequence-based reagent | V391T | CCAGAGTCGAGCAA CTCTAGCAGCCAACG | | PCR primer |
| Sequence-based reagent | K418C | GTACGAACGTCGTGTGT GTAAACGGCGTGCCCGTC | | PCR primer |
| Sequence-based reagent | K577H | AAGCAACAAATTTGTTCT TCACTTGAACTCCGAAACATCC | | PCR primer |
| Sequence-based reagent | TCTN-f | GGAAGATCTATGAAGGAAGTG | | PCR primer |
| Sequence-based reagent | TCTN-r | CCGCTCGAGGCAAAGTTG | | PCR primer |
| Sequence-based reagent | Vang-f | TATGCGGCCGCTCATG GAAAACGAATCCGTC | | PCR primer |
| Sequence-based reagent | Vang-584-r | ATATCTAGATTATAC GGATGTTTCGGAGTT | | PCR primer |
| Sequence-based reagent | Vang-323-r | ATATCTAGATTAGTAG GAGCGCGAAACGCC | | PCR primer |
| Sequence-based reagent | Vang-303-r | ATATCTAGATTAGT GTCGCAGCTCTAGTAA | | PCR primer |
| Commercial assay or kit | Effectene | Qiagen | 301427 | |
| Commercial assay or kit | GFP-Trap Agarose | Chromotek | gta | |
| Software, algorithm | FijiWingsPolarity | (*Dobens et al., 2018*) | | |
| Other | Lambda Protein Phosphatase | NEB | P0753S | |

## Fly strains

Flies were raised on standard medium, and maintained at 25°C unless otherwise stated. To generate *UAS-Vang-Flagx3* transgenic flies, Vang-Flagx3 (*Kelly et al., 2016*) was PCR-amplified using phusion high-fidelity DNA polymerase (Thermo Scientific) and cloned into the pUAST-attB vector (*Bischof et al., 2007*) using NotI-XbaI sites. Point mutants were created using site-directed mutagenesis, the primers used can be found in the key resources table.

Plasmids were verified by Sanger sequencing (GENEWIZ) and sent to BestGene Inc for insertion into BDSC stock number 9752 - *PBAC{yellow[+]-attP-3B}VK00037.*

## S2 culture, pull-downs, and immunoblotting

S2 cells were grown according to standard procedures in Schneider's Medium (Gibco) supplemented with 10% heat-inactivated Fetal Bovine Serum (Gibco). Effectene (QIAGEN) was used to transfect plasmids into S2 cells according to manufacturer's protocols. For details of constructs used please refer to the key resources table. Cells were transfected for ~48 hr before lysis in buffer containing 50 mM Tris-HCl pH 7.5, 150 mM NaCl, 1 mM EDTA and 1% Triton-X-100.

For pull-down experiments with Flag, 10 µl of anti-Flag M2 affinity gel was used per sample (Sigma Aldrich). Lysates were incubated with affinity gel at 4°C, followed by two washes with buffer containing 50 mM Tris-HCl pH 7.5, 350 mM NaCl, 1 mM EDTA and two washes with buffer

containing 50 mM Tris-HCl pH 7.5, 500 mM NaCl, 1 mM EDTA, 0.1% SDS. For GFP pull-down, GFP-Trap agarose was used (Chromotek). In this case, cells were lysed in buffer containing; 10 mM Tris-HCl pH7.5, 150 mM NaCl, 0.5 mM EDTA, and 1% Triton-X-100. This was diluted prior to agarose incubation as per the manufacturer's protocol. Lysates were incubated with 20 µl agarose at 4°C followed by four washes with buffer containing 10 mM Tris-HCl pH7.5, 350 mM NaCl, 0.5 mM EDTA. For both pull-downs, samples were eluted through boiling at 95°C in 5x final sample buffer. Wing discs were dissected from third instar larvae and lysates prepared by boiling collected discs at 95°C in 5x final sample buffer. To perform phosphatase treatment, lysates were exposed to lambda protein phosphatase (NEB) for 30 min at 30°C.

Lysates were resolved by polyacrylamide gel electrophoreses and transferred to nitrocellulose membrane. The primary antibodies and concentration used for immunoblotting can be found in the key resources table. HRP conjugated secondary antibodies were used at 1:5000 (Jackson ImmunoResearch Laboratories).

### *Drosophila* dissection and immunohistochemistry

Adult wings were collected in PBS containing 0.1% Triton-X-100 (PBST) and incubated for 1 hr at room temperature before mounting in 80% glycerol in PBS. Pupal wings were dissected and fixed in 4% paraformaldehyde containing 0.1% Triton-X-100 for 45 mins–1 hr. Tissue was washed with PBST twice and incubated in 5% donkey or goat serum containing PBST for 15 min. Primary antibodies were added and tissue incubated overnight at 4°C. Tissue was washed three times with PBST before incubation with fluorescent secondary antibodies and phalloidin diluted in 5% serum PBST for 2–4 hr at room temperature. Samples were washed four times with PBST and mounted in Vectashield media (Vector Labs). Primary antibodies used are listed in the key resources table. Secondary antibodies were from Jackson ImmunoResearch Laboratories (1:200) and Phalloidin from Molecular Probes (1:1000). Eye sections were prepared as previously described (*Gaengel and Mlodzik, 2008*), and eyes were sectioned near the equatorial region for analysis.

### Imaging, quantification and statistical analysis

Imaging of adult wings and eye sections was performed on a Zeiss Axioplan microscope, imaging of pupal wings was carried out on either a Leica SP5 or Zeiss 880 confocal microscope. Hair reorientation angles were quantified using the FijiWingsPolarity plugin, details of which can be found in *Dobens et al. (2018)*. To perform the quantification, equivalent regions were cropped from three wings in each genotype. Angles of polarity were determined utilizing the plugin and the angles from each genotype combined. For statistical analysis, angles were divided into three categories: anterior reorientation, wild-type, and posterior reorientation. This satisfied the conditions of a valid *Chi*-squared test. Adult eye sections were assigned chirality by hand and orientation angle was defined using ImageJ. For statistical analysis, a *Chi*-squared test was performed.

### Experimental design

In all cases, experiments were performed on at least 3 distinct occasions to ensure technical replication. To ensure biological replication for experiments involving *Drosophila* tissue, 3–10 individual flies were examined for phenotypic similarity. All images and blots within the study were selected as the most representative of the population or findings after this analysis. This approach and sample size is consistent with previous studies performed in our laboratory that have generated reproducible data.

### Statistical analysis

In all cases, unless otherwise stated a Chi-square test was performed. This analysis was performed as it is an accepted test for differences between binned distributions as is the case for our data, and so our analysis examined whether we could disprove to a certain level of significance, the null hypothesis that two data sets are drawn from the same population distribution function. For analysis of actin rotation due to the low values in the medium and severe categories these categories were combined and a Fisher's exact test was performed. The Fisher's exact test was performed in place of the Chi-square test as the low sample size in specific categories meant conditions of the latter would not be

satisfied leading to inaccurate analysis. All analyses were done using Prism software and N value and p values are detailed in tables below.

| Figure 2B - Region 1 | p value | Figure 2B - Region 2 | p value |
| --- | --- | --- | --- |
| ac > WT vs. ac > D317E | <0.0001 | ac > WT vs. ac > D317E | <0.0001 |
| ac > WT vs. ac > R321L | <0.0001 | ac > WT vs. ac > R321L | <0.0001 |
| ac > WT vs. ac > R332H | <0.0001 | ac > WT vs. ac > R332H | <0.0001 |
| ac > WT vs. ac > V391T | 0.1083 | ac > WT vs. ac > V391T | <0.0001 |
| ac > WT vs. ac > K418C | <0.0001 | ac > WT vs. ac > K418C | <0.0001 |
| ac > WT vs. ac > K577H | <0.0001 | ac > WT vs. ac > K577H | <0.0001 |

| Figure 2B | Region 1 N | Region 2 N |
| --- | --- | --- |
| Vang -/- | 188 | 739 |
| W1118 | 118 | 507 |
| WT | 204 | 588 |
| D317E | 111 | 581 |
| R321L | 167 | 609 |
| R332H | 142 | 658 |
| V391T | 157 | 619 |
| K418C | 150 | 538 |
| K577H | 117 | 648 |

| Figure 5D- Chirality | p value | Figure 5D - Rotation | p value |
| --- | --- | --- | --- |
| ac > WT vs. ac > D317E | <0.0001 | ac > WT vs. ac > D317E | <0.0001 |
| ac > WT vs. ac > R321L | 0.0267 | ac > WT vs. ac > R321L | 0.3948 |
| ac > WT vs. ac > R332H | 0.0004 | ac > WT vs. ac > R332H | 0.2055 |
| ac > WT vs. ac > V391T | 0.068 | ac > WT vs. ac > V391T | 0.2481 |
| ac > WT vs. ac > K418C | 0.2935 | ac > WT vs. ac > K418C | 0.0017 |
| ac > WT vs. ac > K577H | <0.0001 | ac > WT vs. ac > K577H | 0.0015 |

| Figure 5D | Chirality - N | Rotation - N |
| --- | --- | --- |
| Vang -/- | 526 | 177 |
| WT | 226 | 146 |
| D317E | 354 | 151 |
| R321L | 491 | 134 |
| R332H | 218 | 123 |
| V391T | 812 | 105 |
| K418C | 667 | 143 |
| K577H | 768 | 171 |

| Figure 5—figure supplement 1B - Chirality | p value | Figure 5—figure supplement 1B - Rotation | p value |
| --- | --- | --- | --- |
| ac > WT vs. ac > D317E | <0.0001 | ac > WT vs. ac > D317E | 0.0003 |
| ac > WT vs. ac > R321L | 0.0225 | ac > WT vs. ac > R321L | 0.0048 |
| ac > WT vs. ac > R332H | 0.1896 | ac > WT vs. ac > R332H | 0.0001 |
| ac > WT vs. ac > V391T | 0.2827 | ac > WT vs. ac > V391T | <0.0001 |

*Continued on next page*

*Continued*

| Figure 5—figure supplement 1B - Chirality | p value | Figure 5—figure supplement 1B - Rotation | p value |
|---|---|---|---|
| ac > WT vs. ac > K418C | 0.1779 | ac > WT vs. ac > K418C | 0.5566 |
| ac > WT vs. ac > K577H | 0.0043 | ac > WT vs. ac > K577H | 0.0267 |

| Figure 5—figure supplement 1B | Chirality - N | Rotation - N |
|---|---|---|
| Vang -/- | 365 | 100 |
| WT | 325 | 89 |
| D317E | 312 | 112 |
| R321L | 203 | 101 |
| R332H | 234 | 125 |
| V391T | 266 | 127 |
| K418C | 428 | 134 |
| K577H | 348 | 118 |

# Acknowledgements

The authors thank members of the Mlodzik lab for helpful suggestions and discussions during the development of the project. We thank Giovanna Collu and Robert Krauss for constructive comments on the manuscript. In particular we are grateful to Neeta Bala, who generated the pTub-Tctn-GFP vector, Jun Wu for the PatJ antibody, and David Strutt (University of Sheffield) for generously providing the *ac-GFP-Vang* transgenic fly strain. We also thank the ISMMS microscopy CoRE, where confocal microscopy was performed and was in part supported by the Tisch Cancer Institute P30 CA196521 grant from the NCI. This research was supported by National Institutes of Health grants R35 GM127103 and R01 EY013256 (to MM) and EMBO (post-doctoral fellowship to ACH).

# Additional information

## Funding

| Funder | Grant reference number | Author |
|---|---|---|
| National Institute of General Medical Sciences | R35 GM127103 | Marek Mlodzik |
| National Eye Institute | R01 EY013256 | Marek Mlodzik |
| European Molecular Biology Organization | | Ashley C Humphries |

The funders had no role in study design, data collection and interpretation, or the decision to submit the work for publication.

## Author contributions

Ashley C Humphries, Conceptualization, Data curation, Formal analysis, Validation, Investigation, Visualization, Methodology, Writing - original draft, Writing - review and editing; Sonali Narang, Investigation, Writing - review and editing; Marek Mlodzik, Conceptualization, Formal analysis, Supervision, Funding acquisition, Investigation, Writing - original draft, Project administration, Writing - review and editing

## Author ORCIDs

Ashley C Humphries (iD) https://orcid.org/0000-0002-0737-801X
Marek Mlodzik (iD) https://orcid.org/0000-0002-0628-3465

Decision letter and Author response

Decision letter https://doi.org/10.7554/eLife.53532.sa1

Author response https://doi.org/10.7554/eLife.53532.sa2

## Additional files

### Supplementary files

• Transparent reporting form

### Data availability

All data generated or analysed during this study are included in the manuscript and supporting files. Source data files have been provided for Figures 1, 2 and 5.

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
