## [Decision Letter]

**Acceptance summary:**

By modeling human and mice mutations of Van Gogh, a planar cell polarity gene, in *Drosophila*, your work provides interesting mechanistic data into how these variants contribute to neural tube defects.

**Decision letter after peer review:**

Thank you for submitting your article "Mutations associated with human neural tube defects display disrupted planar cell polarity in *Drosophila*" for consideration by *eLife*. Your article has been reviewed by Marianne Bronner as the Senior Editor, Hugo Bellen as the Reviewing Editor, and three reviewers. We apologize for the lengthy review process but the holidays did not help. The reviewers have opted to remain anonymous.

The reviewers have discussed the reviews with one another and the Reviewing Editor has drafted this decision to help you prepare a revised submission. Please address the concerns of reviewer one and where possible those of reviewers 2 and 3.

We are enclosing the unedited version of the three reviewers comments, as they had some disagreements between them to which we wanted you to be privy. However, after discussion, the reviewers agreed that your paper is suitable for publication in *eLife* but that it will require extensive editing to make it more reader friendly for a broad audience (rather than just Drosophilists), and that it is better polished. Note that in the future, a more sophisticated approach should be envisaged using CRISPR technology and that point mutations in the endogenous gene will become the norm, rather than overexpression studies.

Reviewer #1:

I quite like this new paper, which attacks the knotty problem of assessing pathogenicity within the rapidly expanding universe of disease associated variants from human patients. NTDs are an important problem, and the PCP genes are well known to be associated with NTDs. However, for the vast majority of variants we have little information relating genotype to phenotype. Here, the authors exploit the unique advantages of the *Drosophila* system to assess the pathogenicity of several variants in Vangl. The paper is very important, then, for developing a new system in which such pathogenicity can be assessed in the biologically relevant setting of a planar polarized epithelium. It therefore lays down the groundwork for systematic assessment of PCP variants in human disease.

That said, the paper suffers from being inaccessible to non-Drosopholists and for providing too little context for the findings. With more effort put into the writing, this paper would be an excellent fit for *eLife*.

Comments, in order of appearance in the manuscript:

1) Improved scholarship would help: The significance of this paper rests in the synthesis of ideas and the integration of scientific communities (i.e. It exploits model system (fly) genetics to understand human disease genetics). To be effective, then, the paper must not alienate any of the stakeholders. So, the biggest liability is the paper's failure to meaningfully deal with the literature outside of the fly. This must be corrected:

Introduction. It's not clear why the Kibar 2007 paper is cited here. It is not a review of NTDs, but rather a primary paper showing for the first time the Vangl2 is mutated in human NTDs. It would more properly be cited in passages linking Vangl2 to NTDs (i.e. later in the Introduction).

Introduction. Several errors: Tada and Smith, 2000 did not look at neutral tube closure or neural tissue at all. It should be removed. Wallingford et al., 2002 is a review of PCP and convergent extension and does not mention neural tube closure in any depth; this should be replaced with the relevant paper for the link to neural plate width, which is Wallingford and Harland, 2002. Likewise, while the Wang, 2006b reference does describe PCP genes and neural tube closure, Wang, 2006a is the paper that actually links PCP to neural plate width. It should be cited here.

Introduction: Wang, 2006b should be included in the list here.

Introduction: The Looptail mouse was studied by geneticists for decades before it was identified as a mutation in Vangl2. That work is notable because the odd penetrance of the allele led to the conclusion that the mutation was "semi-dominant," a decades-old conclusion confirmed by the nice experiments here. I recommend the authors discuss and cite (at a minimum) the following:

Developmental basis of severe neural tube defects in the loop-tail (Lp) mutant mouse: use of microsatellite DNA markers to identify embryonic genotype.

Copp, Checiu and Henson, 1994.

All-or-none craniorachischisis in Loop-tail mutant mouse chimeras.

Musci and Mullen, 1990.

Strong and Hollander, (1949).

These papers might also be re-introduced in the Discussion section.

The Introduction almost seems to be specifically designed to irritate non-Drosophilists. One might read a passage like this and retort that the linkage of PCP to NTDs is really the only truly important PCP finding, and it was made in the mouse. Why risk it?

2) Not written accessibly: I was able eventually to figure out what was what in Figure 2, but this critical figure was very poorly described.

Subsection “*Drosophila* provides a sensitive system for investigating the effect of mutations on PCP”, I don't know know what "attP/B sites" are, would nice to have this system explained.

Figure 2. I do not know what *w1118* is. After some digging, I find it is the control. But then, why is it shown after the mutant? What are we meant to compare the others to? The mutant or the control?

Figure 2. "ac>" is never explicitly defined, though I gather it means the actin-Gal4 driven construct.

Figure 2. All the text in B is unreadably small.

Subsection “*Drosophila* provides a sensitive system for investigating the effect of mutations on PCP”: The text says "all showed a notable change in protein mobility, but the data in Figure 2C do not convince me. What changes are they claiming? Is it a shift? A loss of a second band?

Subsection “*Drosophila* provides a sensitive system for investigating the effect of mutations on PCP”: I do not think it's appropriate to speculate these shifts may reflect phosphorylation here. Either do the simple experiment or move this to the Discussion section.

3) Overall lack of clarity: The experiments here are good ones, and overall, I find them compelling. Just the same, they are complex and must be explained clearly. Confusing and even misleading statements include:

Subsection “Dominant behavior and protein localization defects in the majority of mutations”: "….comparison of localization of "endogenous wt-Vang…."" The interior set of quotation marks here are the authors, indicating that even they know this experiment has nothing to do with actual endogenous vangl. The word endogenous must be removed and the authors must make it clear that this experiment looks at the effect of over-expressed mutant vangl on co-overexpressed mutant vangl.

Subsection “Dominant behavior and protein localization defects in the majority of mutations”: "…suggestive of competition from the over-expressed Vang-flag for membrane recruitment. I concur with the authors' interpretation, but this result contradicts their claim that their overexpressions "remain within physiological levels". This should be stated and addressed.

Subsection “Mutants show altered membrane localization in rescue conditions”: "as compared to wt-Vang." Unless I am mistaken, this should be corrected to indicate that it is as compared to OVEREXPRESSED wt-Vang.

4) Subsection “Mutations associated with reduced membrane localization or loss in protein function disrupt effector binding”. An alternative interpretation is that these mutations lead to trapping of Vang in the ER or Golgi, where it cannot access cytoplasmic proteins such as Dvl, thus leading to the observed loss of binding. This is known to be the case for D317E, so this possibility should be raised here.

5) I think it would be great to include a table that summarizes the data for each mutation and ALSO summarizes the associated human phenotype (i.e. the specifics of the NTD for each).

Reviewer #2:

Humphries et al., report the most comprehensive to date in vivo study in the *Drosophila* model of neural tube defect (NTD) associate mutations in VANGL2 and VANGL1, the core (and as the authors argue the most "specific) component of the Planar Cell Polarity pathway. A subset of six missense mutations associated with NTDs in human and mouse in the C-terminal region of VANGL is studied. The authors leverage the advantage of the *Drosophila* model, in which the PCP pathway has been discovered and is arguably best understood. Using Gal4/UAS overexpression system they express the *Drosophila* Flagx3-tagged Vangl protein with the NTD mutations in the equivalent amino acid residues in wild type (overexpression) and vangl2 mutant background (rescue) and quantifying the wing hair orientation, as well as intracellular/tissue distribution of the (tagged) mutant and endogenous Vangl2 proteins, they conclude that the investigated candidate mutations are causative, and some work as antimorphs, hypermorphs, and one as a hypomorph. They identify reduced membrane localization, reduced interaction with intracellular effectors such ad Dsh, Pk, and Scribble and the interference with the membrane localization of the endogenous Vang as key molecular defects of the mutant proteins. One interesting insight is that "these mutations do not show specificity for a particular effector but instead a general reduction in binding" what the authors interpret as an effect of the mutations on "the integrity of the entire protein".

Whereas, some studies of VANGL mutations have been reported in other systems (zebrafish, mammalian cell culture), this is the most comprehensive in vivo study. The manuscript reports a large amount of data, is well-reasoned and clearly presented. The authors carefully assess also the level of the analyzed proteins, their mobility in gel electrophoresis, and intracellular distribution. In combination with functional studies this allows them to deduce how individual mutations affect function of the various mutant proteins in PCP signaling in vivo. This confirms and significantly extends some of the previous work (e.g. on the hypomorphic nature of the Vangl2Lp424 -R259L mutation). As such the study should be of interest to the scientific community. However, there are several questions about the interpretations and limitations of the experimental approach that need to be addressed before the manuscript is suitable for publication.

The study is a missed opportunity, because the current experimental approach has limitations as discussed below. Indeed, it is somewhat surprising that in times when genome editing in *Drosophila* is feasible, the author chose ectopic overexpression rather than engineering these mutations into the endogenous locus.

The limitation of the current overexpression approach is best seen in the rescue experiments. Here, the authors used actin-Gal4 driven expression, which as the authors state "due to the higher than endogenous expression levels of the transgenes, we observed a gain-of-function phenotype with wt-Vang (Figure 4A). Despite this effect, we were able to observe hair reorientation patterns in the absence of endogenous Vang that were consistent with the experiments described above". As planar cell polarity and processes regulated by it are exquisitely sensitive to the level of Vangl expression (and other PCP proteins), it has been difficult to rescue Vang/Vangl mutant phenotypes by ectopic expression in any system. This also confounds the interpretation of the results of these experiments.

In the rescue experiments in the eye, e.g. Figure 5 it appears that even the proposed dominant negative mutant forms provide some rescuing activity. Could the authors comment on this?

On the other hand, the rescue experiments in the wing, do not allow sufficient resolution to ask whether the cell polarity phenotypes in the wing of vang-/- mutants overexpressing the proposed dominant negative Vang mutant proteins are identical to vang mutants, or stronger, or possibly different. This would allow to distinguish between the antimorphic activity of these proteins being exerted through their effect on endogenous Vang protein (as seen in overexpression experiments) or possibly on other proteins as well.

Reviewer #3:

In this manuscript, Humphries et al. use *Drosophila* to examine several mammalian Vangl1/2 mutations that have been associated with neural tube defects. Equivalent mutations were made in *Drosophila* Vang in overexpression systems, with all versions FLAG tagged and placed into attP/B sites for equivalent transcript expression. D317E, R321L, R332H, V391T, K418C and K577H mutations were examined. Overexpression of wt-Vang leads to consistent patterns of hair reorientation in the wing, and the authors found that overexpression of different Vangl mutations generally yields different patterns. They also found that a subset of the mutations have different protein mobility, consistent with the possibility that phosphorylation was altered, although phosphatase treatment was not used to confirm this. Staining revealed some of the mutations have altered localization while others had normal membrane localization. These data led the authors to conclude the D317E and K577H are dominant-negative alleles, and disrupt the localization of wildtype Vang, R332H and K418C are dominant mutations which remain at the membrane, R321L is a hypomorph and V391T is a mild GOF allele. The authors also expressed these alleles in the eye in Vangl mutants, using actin Gal4 and sep-Gal4. Binding to Dsh and Prickle was tested for all the alleles in S2 cells, and decreased interactions were noted for a subset of the alleles (317,321,577). The authors also used pull-down experiments to determine the Dgo binding to Vang requires Vang 303-322.

The experiments are well described, the data are well presented, and a significant amount of work was put into generating and analyzing all of the alleles. This study supports the proposal that the human-derived mutations cause PCP defects. Largely the work also supports previous studies in mouse models, and studies in other systems (MDCK cells and zebrafish) of Vangl alleles, although certain alleles act differently in the *Drosophila* system.

However, while this study supports the proposal that *Drosophila* can be used to model human mutations to probe for causality, the paper does not lead to significant new insights into how planar cell polarity is established, nor why these mutations lead to NTD. Ideally these mutations would be made in the endogenous locus, using CRISPR, and the presence of wildtype Vang complicates the interpretation in the wing. In addition, the high levels of expression precluded the authors examining PCP localization of their alleles. Overall, this paper seems more suited to a journal such as Genetics.

---

## [Author Response]

We are enclosing the unedited version of the three reviewers comments, as they had some disagreements between them to which we wanted you to be privy. However, after discussion, the reviewers agreed that your paper is suitable for publication in eLife but that it will require extensive editing to make it more reader friendly for a broad audience (rather than just Drosophilists), and that it is better polished. Note that in the future, a more sophisticated approach should be envisaged using CRISPR technology and that point mutations in the endogenous gene will become the norm, rather than overexpression studies.Reviewer #1:I quite like this new paper, which attacks the knotty problem of assessing pathogenicity within the rapidly expanding universe of disease associated variants from human patients. NTDs are an important problem, and the PCP genes are well known to be associated with NTDs. However, for the vast majority of variants we have little information relating genotype to phenotype. Here, the authors exploit the unique advantages of the *Drosophila* system to assess the pathogenicity of several variants in Vangl. The paper is very important, then, for developing a new system in which such pathogenicity can be assessed in the biologically relevant setting of a planar polarized epithelium. It therefore lays down the groundwork for systematic assessment of PCP variants in human disease.

We thank the reviewer for their kind comments and are glad they liked our study.

That said, the paper suffers from being inaccessible to non-Drosopholists and for providing too little context for the findings. With more effort put into the writing, this paper would be an excellent fit for eLife.

Our apologies, we have worked to address this and are grateful to the reviewer for pointing out the areas of greatest concern. We have also used a FijiWingsPolarity software based presentation to replace the schematic drawings of wings within Figure 3 and Figure 4. This is in order to promote consistency with our analysis in Figure 2 and also to allow readers an easier interpretation of the different phenotypic features and patterns. We have moved the schematic drawings to the supplement, so we have two ways of representing the data, again hopefully improving accessibility for a wider audience.

Essential revisions:1) Improved scholarship would help: The significance of this paper rests in the synthesis of ideas and the integration of scientific communities (i.e. It exploits model system (fly) genetics to understand human disease genetics). To be effective, then, the paper must not alienate any of the stakeholders. So, the biggest liability is the paper's failure to meaningfully deal with the literature outside of the fly. This must be corrected:Introduction. It's not clear why the Kibar 2007 paper is cited here. It is not a review of NTDs, but rather a primary paper showing for the first time the Vangl2 is mutated in human NTDs. It would more properly be cited in passages linking Vangl2 to NTDs (i.e. later in the Introduction).

Thank you for pointing this out, we have removed the citation. As we have cited a comprehensive review regarding the link between Vangl and human NTD in the introduction we have chosen to cite the Kibar, 2007 when directly referring to the findings of the paper e.g. Introduction, subsection “Mutations associated with mammalian NTD can be found throughout the C-terminal tail of *Vangl* genes”.

Introduction. Several errors: Tada and Smith, 2000 did not look at neutral tube closure or neural tissue at all. It should be removed. Wallingford et al., 2002 is a review of PCP and convergent extension and does not mention neural tube closure in any depth; this should be replaced with the relevant paper for the link to neural plate width, which is Wallingford and Harland, 2002. Likewise, while the Wang, 2006b reference does describe PCP genes and neural tube closure, Wang, 2006a is the paper that actually links PCP to neural plate width. It should be cited here.

Our apologies we made a mistake and referenced the wrong Wallingford and Wang papers, thank you for alerting us to this. We have corrected this error.

Introduction: Wang, 2006b should be included in the list here.

We have added this reference, apologies for the initial omission.

Introduction: The Looptail mouse was studied by geneticists for decades before it was identified as a mutation in Vangl2. That work is notable because the odd penetrance of the allele led to the conclusion that the mutation was "semi-dominant," a decades-old conclusion confirmed by the nice experiments here. I recommend the authors discuss and cite (at a minimum) the following:Developmental basis of severe neural tube defects in the loop-tail (Lp) mutant mouse: use of microsatellite DNA markers to identify embryonic genotype.Copp, Checiu and Henson, 1994.All-or-none craniorachischisis in Loop-tail mutant mouse chimeras.Musci and Mullen, 1990.Strong and Hollander, (1949).These papers might also be re-introduced in the Discussion section.Thank you for pointing this out. We have included these references and added more discussion surrounding the original *Lp* mouse work, e.g. lines Introduction and subsection “Functional definition of both dominant negative and gain-of-function mutations”.The Introduction almost seems to be specifically designed to irritate non-Drosophilists. One might read a passage like this and retort that the linkage of PCP to NTDs is really the only truly important PCP finding, and it was made in the mouse. Why risk it?

This sentence has been removed to reflect the reviewer’s concerns. Our intention was only to highlight the advantages of using the *Drosophila* system for certain types of analysis due to its simplicity compared to mammalian systems, which we discuss later in the text.

2): Not written accessibly: I was able eventually to figure out what was what in Figure 2, but this critical figure was very poorly described.Subsection “*Drosophila* provides a sensitive system for investigating the effect of mutations on PCP”, I don't know know what "attP/B sites" are, would nice to have this system explained.

Apologies for not having explained this clearly. We have added text to better explain the attP/B system that allows direct comparison of individual transgenes as they’re all expressed from the same genomic insertion. Please see subsection “*Drosophila* provides a sensitive system for investigating the effect of mutations on PCP”.

Figure 2. I do not know what w1118 is. After some digging, I find it is the control. But then, why is it shown after the mutant? What are we meant to compare the others to? The mutant or the control?

The *w1118* genotype is used as a reference control where no phenotype is observed (“wild-type”). We have reordered the figure to have this “wild-type” reference control first. We describe now in the main text that *w1118* is genetically wild-type, and we have now added more text to hopefully make clearer to a wider audience what this fly genotype is and why we include it. Please see subsection “*Drosophila* provides a sensitive system for investigating the effect of mutations on PCP” and Figure 2 legend.

The different phenotypes can and should be compared to the reference control, the loss-of-function, and the wt overexpression. This is most useful, as the different mammalian mutants show different hair-reorientation patterns that are reminiscent of the three aforementioned genotypes. These different patterns and phenotypes are discussed within the text, e.g. subsection “*Drosophila* provides a sensitive system for investigating the effect of mutations on PCP”. For statistical analysis we compared the wt overexpression to each mutant overexpression, as we feel this is the most appropriate comparison and control.

Figure 2. "ac>" is never explicitly defined, though I gather it means the actin-Gal4 driven construct.

Thank you for pointing this out. It is has now been introduced both in the main text and in the figure legend. subsection “*Drosophila* provides a sensitive system for investigating the effect of mutations on PCP” and Figure 2 legend.

Figure 2. All the text in B is unreadably small.We apologize for this mishap, we have increased the text size in the figure.Subsection “*Drosophila* provides a sensitive system for investigating the effect of mutations on PCP”: The text says "all showed a notable change in protein mobility, but the data in Figure 2C do not convince me. What changes are they claiming? Is it a shift? A loss of a second band?There are a number of papers detailing the phosphorylation of Vang(l) proteins and associated changes in protein mobility as referenced in the manuscript. This includes our labs previous study (Kelly et al., 2016) as well as other *Drosophila* work (Strutt et al., 2019). Please note the variation in appearance of the phospho band shifts here. We find that loss in phosphorylation can lead to a single lower band or collapsed bands depending on the gel. To further clarify this point, we have repeated these experiments to show the change in protein mobility more consistently throughout the manuscript and the new data can be seen in Figure 2—figure supplement 1C.Subsection “*Drosophila* provides a sensitive system for investigating the effect of mutations on PCP”: I do not think it's appropriate to speculate these shifts may reflect phosphorylation here. Either do the simple experiment or move this to the Discussion section.

We have performed the respective experiments with phosphatase (PPase) treatment, the new data set directly comparing controls and PPase treated samples can be seen in the new Figure 2—figure supplement 1C.

3) Overall lack of clarity: The experiments here are good ones, and overall, I find them compelling. Just the same, they are complex and must be explained clearly. Confusing and even misleading statements include:Subsection “Dominant behavior and protein localization defects in the majority of mutations”: "….comparison of localization of "endogenous wt-Vang…."" The interior set of quotation marks here are the authors, indicating that even they know this experiment has nothing to do with actual endogenous vangl. The word endogenous must be removed and the authors must make it clear that this experiment looks at the effect of over-expressed mutant vangl on co-overexpressed mutant vangl.

We apologize for the misleading comment, it was not our intention. We have added to the text to make the experimental approach easier to understand. Please see subsection “Dominant behavior and protein localization defects in the majority of mutations” in the revised manuscript.

Subsection “Dominant behavior and protein localization defects in the majority of mutations”: "…suggestive of competition from the over-expressed Vang-flag for membrane recruitment. I concur with the authors' interpretation, but this result contradicts their claim that their overexpressions "remain within physiological levels". This should be stated and addressed.

We appreciate the comment and concern of the reviewer, however, we disagree that there is a contradiction. In this case the Vang-flag is overexpressed, while Vang-GFP remains within physiological levels, as shown by the absence of any GOF phenotypic change by the Vang-GFP expression (Figure 3—figure supplement 1A). The change in Vang-GFP localization is due to the abundance of Vang-flag within the cell. We believe that better introduction of the experiment has made clearer the difference in expression levels between the Vang-flag mutants and Vang-GFP.

Subsection “Mutants show altered membrane localization in rescue conditions”: "as compared to wt-Vang." Unless I am mistaken, this should be corrected to indicate that it is as compared to OVEREXPRESSED wt-Vang.

Thank you for pointing this out. Yes, this has been corrected.

4) Subsection “Mutations associated with reduced membrane localization or loss in protein function disrupt effector binding”. An alternative interpretation is that these mutations lead to trapping of Vang in the ER or Golgi, where it cannot access cytoplasmic proteins such as Dvl, thus leading to the observed loss of binding. This is known to be the case for D317E, so this possibility should be raised here.

We have discussed this interpretation in the Discussion section. We feel the data from the *Lp* mouse more likely reflect a loss of binding due to altered structure rather than trapping away from effectors. This is due to the observation that Sec24b is unable to bind and this is the reason for its localization in the ER (discussed in subsection “Functional definition of both dominant negative and gain-of-function mutations”). We also think that there is not strong enough evidence to show that the D317E and K577H mutants are trapped within a particular sub compartment, so we are reluctant to state this within the text. We do however discuss the trapping of *Lp* in the ER, so that readers are aware of these findings (subsection “*Drosophila* as model to define causative nature of human mutations”). Finally, we know that R321L can be membrane localized and so this mutant would be able to access cytoplasmic proteins and it still shows a binding defect.

5) I think it would be great to include a table that summarizes the data for each mutation and ALSO summarizes the associated human phenotype (i.e. the specifics of the NTD for each).Thank you for the suggestion. We agree that it would be nice to correlate the human data directly with our work. Unfortunately, we feel that this is not possible. As NTDs are very complex, it is clear that a number of factors combine to contribute to the overall phenotype (as discussed in our manuscript, Introduction). We therefore feel that stating the phenotype of one particular patient would be an oversimplification, as we cannot be sure whether and what other factors are contributing to disease progression, this is also relevant when considering that some mutations are familial and family members show different phenotypic outcome. We have thus chosen to keep our summary table as in the original version, however, should the editors feel differently on this point, it would of course be possible to adjust this and add this information.Reviewer #2:Humphries et al., report the most comprehensive to date in vivo study in the Drosophila model of neural tube defect (NTD) associate mutations in VANGL2 and VANGL1, the core (and as the authors argue the most "specific) component of the Planar Cell Polarity pathway. A subset of six missense mutations associated with NTDs in human and mouse in the C-terminal region of VANGL is studied. The authors leverage the advantage of the *Drosophila* model, in which the PCP pathway has been discovered and is arguably best understood. Using Gal4/UAS overexpression system they express the *Drosophila* Flagx3-tagged Vangl protein with the NTD mutations in the equivalent amino acid residues in wild type (overexpression) and vangl2 mutant background (rescue) and quantifying the wing hair orientation, as well as intracellular/tissue distribution of the (tagged) mutant and endogenous Vangl2 proteins, they conclude that the investigated candidate mutations are causative, and some work as antimorphs, hypermorphs, and one as a hypomorph. They identify reduced membrane localization, reduced interaction with intracellular effectors such ad Dsh, Pk, and Scribble and the interference with the membrane localization of the endogenous Vang as key molecular defects of the mutant proteins. One interesting insight is that "these mutations do not show specificity for a particular effector but instead a general reduction in binding" what the authors interpret as an effect of the mutations on "the integrity of the entire protein".Whereas, some studies of VANGL mutations have been reported in other systems (zebrafish, mammalian cell culture), this is the most comprehensive in vivo study. The manuscript reports a large amount of data, is well-reasoned and clearly presented. The authors carefully assess also the level of the analyzed proteins, their mobility in gel electrophoresis, and intracellular distribution. In combination with functional studies this allows them to deduce how individual mutations affect function of the various mutant proteins in PCP signaling in vivo. This confirms and significantly extends some of the previous work (e.g. on the hypomorphic nature of the Vangl2Lp424 -R259L mutation). As such the study should be of interest to the scientific community. However, there are several questions about the interpretations and limitations of the experimental approach that need to be addressed before the manuscript is suitable for publication.We thank the reviewer for their positive comments surrounding the findings of our study.The study is a missed opportunity, because the current experimental approach has limitations as discussed below. Indeed, it is somewhat surprising that in times when genome editing in Drosophila is feasible, the author chose ectopic overexpression rather than engineering these mutations into the endogenous locus.

We appreciate that these experiments would of have been an excellent addition, however, we chose the UAS/Gal4 attP/B system to be able to manipulate levels and dynamics of expression to answer different questions. We have discussed our reasoning for our approach in the cover letter, and for completeness we include our comments again here:

We were particularly interested to explore the nature of the mutations, e.g. gain of function or loss of function. Therefore, we felt the overexpression experiment would nicely address this, shown by either an enhancement, reduction or no change in phenotype as compared to wt overexpression. This coupled with the rescue experiment could then delineate between hypermorphic, hypomorphic, null or dominant negative mutations. As the Vang heterozygote displays very little phenotype, we were concerned that introduction of the mutation in the endogenous locus would not yield as clear results, especially for weaker mutations. This is also complicated by the fact that loss-of-function and gain-of-function mutations in core PCP genes both lead to non-polar, abnormal localization. The mutations appear in heterozygote in human patients, however, in these cases the background must be favorable to see pathogenesis (family members with the mutation do not always show disease progression). While we feel we address what we set out to study, showing that mutations found in the human patients do show PCP defects, the next step of course would be to study the intricacies of PCP signaling in an endogenous setting.

The limitation of the current overexpression approach is best seen in the rescue experiments. Here, the authors used actin-Gal4 driven expression, which as the authors state "due to the higher than endogenous expression levels of the transgenes, we observed a gain-of-function phenotype with wt-Vang (Figure 4A). Despite this effect, we were able to observe hair reorientation patterns in the absence of endogenous Vang that were consistent with the experiments described above". As planar cell polarity and processes regulated by it are exquisitely sensitive to the level of Vangl expression (and other PCP proteins), it has been difficult to rescue Vang/Vangl mutant phenotypes by ectopic expression in any system. This also confounds the interpretation of the results of these experiments.In the rescue experiments in the eye, e.g. Figure 5 it appears that even the proposed dominant negative mutant forms provide some rescuing activity. Could the authors comment on this?

We of course agree that there is a limitation to this approach, however we included the rescue experiments to support our findings from the dominant assays (overexpression – phenotype and localization). We also feel that they allowed for further interpretation. For example, without these experiments we would have not been able to confirm that R321L is a hypomorph and not a null allele. While the rescue is not perfect, as we acknowledge within the text, we think that the data is fully consistent with our other results. In regards to the dominant negative rescue, we indeed believe that some function of Vang is retained in these mutants that enables the very minor rescue that we see. We have now discussed this in the manuscript to clarify these points (subsection “Rescue experiments in the eye confirm loss-of-function and gain-of-function behavior).

On the other hand, the rescue experiments in the wing, do not allow sufficient resolution to ask whether the cell polarity phenotypes in the wing of vang-/- mutants overexpressing the proposed dominant negative Vang mutant proteins are identical to vang mutants, or stronger, or possibly different. This would allow to distinguish between the antimorphic activity of these proteins being exerted through their effect on endogenous Vang protein (as seen in overexpression experiments) or possibly on other proteins as well.

We appreciate that the reviewer raises an interesting point. We fully agree and would be keen to follow up the D317E and K577H mutants in a future study, to see whether their dominance impacts only Vang or indeed other PCP effectors. These findings could also have interesting implications for the *Lp* mouse.

Reviewer #3:In this manuscript, Humphries et al. use *Drosophila* to examine several mammalian Vangl1/2 mutations that have been associated with neural tube defects. Equivalent mutations were made in *Drosophila* Vang in overexpression systems, with all versions FLAG tagged and placed into attP/B sites for equivalent transcript expression. D317E, R321L, R332H, V391T, K418C and K577H mutations were examined. Overexpression of wt-Vang leads to consistent patterns of hair reorientation in the wing, and the authors found that overexpression of different Vangl mutations generally yields different patterns. They also found that a subset of the mutations have different protein mobility, consistent with the possibility that phosphorylation was altered, although phosphatase treatment was not used to confirm this. Staining revealed some of the mutations have altered localization while others had normal membrane localization. These data led the authors to conclude the D317E and K577H are dominant-negative alleles, and disrupt the localization of wildtype Vang, R332H and K418C are dominant mutations which remain at the membrane, R321L is a hypomorph and V391T is a mild GOF allele. The authors also expressed these alleles in the eye in Vangl mutants, using actin Gal4 and sep-Gal4. Binding to Dsh and Prickle was tested for all the alleles in S2 cells, and decreased interactions were noted for a subset of the alleles (317,321,577). The authors also used pull-down experiments to determine the Dgo binding to Vang requires Vang 303-322.The experiments are well described, the data are well presented, and a significant amount of work was put into generating and analyzing all of the alleles. This study supports the proposal that the human-derived mutations cause PCP defects. Largely the work also supports previous studies in mouse models, and studies in other systems (MDCK cells and zebrafish) of Vangl alleles, although certain alleles act differently in the *Drosophila* system.We thank the reviewer for appreciating the effort we have taken with our study, and are glad they found the manuscript and study to be well executed.However, while this study supports the proposal that *Drosophila* can be used to model human mutations to probe for causality, the paper does not lead to significant new insights into how planar cell polarity is established, nor why these mutations lead to NTD. Ideally these mutations would be made in the endogenous locus, using CRISPR, and the presence of wildtype Vang complicates the interpretation in the wing. In addition, the high levels of expression precluded the authors examining PCP localization of their alleles. Overall, this paper seems more suited to a journal such as Genetics.

We very much appreciate the comment of the reviewer and we have commented on the reason for our approach in the cover letter and above (please see also below), and of course we agree that next steps would be to introduce the mutations into the endogenous locus. This would then allow for further analysis of how PCP signaling itself is altered in the mutants and insight into PCP establishment mechanisms. However, this was not the goal of our study, in fact we feel that our work here now validates the utility of such an approach (subsection “*Drosophila* as model to define causative nature of human mutations”).

In response to “using CRIPR to introduce the mutations in the endogenous locus”:

We were particularly interested to explore the nature of the mutations, e.g. gain of function or loss of function. Therefore, we felt the overexpression experiment would nicely address this, shown by either an enhancement, reduction or no change in phenotype as compared to wt overexpression. This coupled with the rescue experiment could then delineate between hypermorphic, hypomorphic, null or dominant negative mutations. As the Vang heterozygote displays very little phenotype, we were concerned that introduction of the mutation in the endogenous locus would not yield as clear results, especially for weaker mutations. This is also complicated by the fact that loss-of-function and gain-of-function mutations in core PCP genes both lead to non-polar, abnormal localization. The mutations appear in heterozygote in human patients, however, in these cases the background must be favorable to see pathogenesis (family members with the mutation do not always show disease progression). While we feel we address what we set out to study, showing that mutations found in the human patients do show PCP defects, the next step of course would be to study the intricacies of PCP signaling in an endogenous setting.

In the case of the majority of these mutants, there was no evidence to suggest that they could impact Vang function. Therefore, we feel that our study importantly shows this, and thus defects in PCP signaling can be causative for NTD in patients, which we feel is a significant addition to the field.